# How Well Does Self-Supervised Pre-Training Perform with Streaming Data?

**Dapeng Hu**[*1] **Shipeng Yan**[*2] **Qizhengqiu Lu**[3] **Lanqing Hong**[4] **Hailin Hu**[3]
**Yifan Zhang**[1] **Zhenguo Li**[4] **Xinchao Wang**[1] **Jiashi Feng**[1]

[1]National University of Singapore     [2]ShanghaiTech University
[3]AARC, Huawei Technologies     [4]Huawei Noah's Ark Lab

## ABSTRACT

Prior works on self-supervised pre-training focus on the joint training scenario, where massive unlabeled data are assumed to be given as input all at once, and only then is a learner trained. Unfortunately, such a problem setting is often impractical if not infeasible since many real-world tasks rely on sequential learning, e.g., data are decentralized or collected in a streaming fashion. In this paper, we conduct the first thorough and dedicated investigation on self-supervised pre-training with streaming data, aiming to shed light on the model behavior under this overlooked setup. Specifically, we pre-train over 500 models on four categories of pre-training streaming data from ImageNet and DomainNet and evaluate them on three types of downstream tasks and 12 different downstream datasets. Our studies show that, somehow beyond our expectation, with simple data replay or parameter regularization, sequential self-supervised pre-training turns out to be an efficient alternative for joint pre-training, as the performances of the former are mostly on par with those of the latter. Moreover, catastrophic forgetting, a common issue in sequential supervised learning, is much alleviated in sequential self-supervised learning (SSL), which is well justified through our comprehensive empirical analysis on representations and the sharpness of minima in the loss landscape. Our findings, therefore, suggest that, in practice, for SSL, the cumbersome joint training can be replaced mainly by sequential learning, which in turn enables a much broader spectrum of potential application scenarios.

## 1 INTRODUCTION

Recent advances in self-supervised learning (SSL) (He et al., 2020; Grill et al., 2020; Caron et al., 2020; Jure et al., 2021) demonstrate competitive or even better transfer learning performance on various downstream tasks, compared with supervised pre-training. Although waiving the cost of human labeling, SSL usually requires massive unlabeled data to learn a powerful representation model and benefits from significantly large-scale pre-training data, e.g., He et al. (2020) adopted billion-scale data to pre-train better SSL models. The common pre-training practice follows the **joint training** (JT) setup, where massive unlabeled data are collected together before model training. In reality, however, it is usually difficult to access a large amount of collective unlabeled data at once. Instead, real-world data are usually accessed in a streaming fashion, e.g., data are generated and collected sequentially chunk by chunk (Delange et al., 2021), or even decentralized and stored in different servers (Lange et al., 2020); such a learning setup is known as **sequential training** (ST). Despite much research effort and promising results achieved by JT, it inevitably suffers from heavy data storage, prolonged training time, and finds itself incompetent when training data volume expands over time. For ST, on the other hand, a learner can be sequentially trained with disjoint data chunks, making it much more efficient than JT.

How to effectively and efficiently pre-train a representation model under the ST setup has been an open problem. Despite the high efficiency, some continual learning research works (Goodfellow et al., 2013; Kirkpatrick et al., 2017; Rebuffi et al., 2017) have shown that ST with supervised models tends to suffer from catastrophic forgetting (McCloskey & Cohen, 1989), having a significant

---

*Contributed equally: `dapeng.hu@u.nus.edu` and `yanshp@shanghaitech.edu.cn`.

performance degradation on the historical data chunks. Unlike the case of continual learning tasks, in pre-training tasks, one expects the model to well generalize to downstream tasks rather than focusing only on the seen tasks (Chen et al., 2020a). Nevertheless, how well sequential self-supervised models perform on downstream tasks remains unclear.

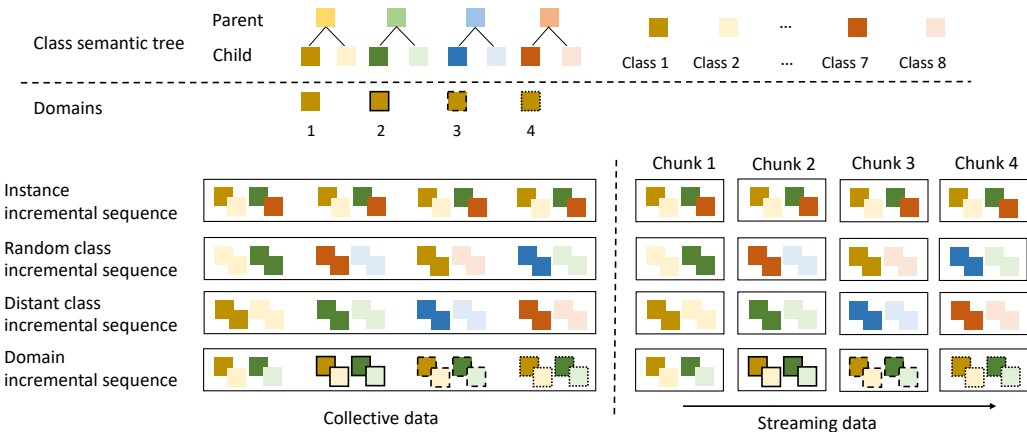

Figure 1: Illustration of streaming data and the corresponding collective data. Different colors denote different classes, and border types distinguish different domains. We use the WordNet Tree (Miller, 1998) to measure the semantic similarity of classes. Classes having the same parent or ancestor in WordNet, marked with similar colors, share similar semantics in the class semantic tree.

To fill the research gap, we provide a thorough empirical study on self-supervised pre-training with streaming data. In the pre-training stage, to mimic real-world data collection scenarios and for the better dissection of sequential SSL, we consider streaming data with different degrees of distribution shifts. As shown in Figure 1, we obtain four types of streaming data, including the instance incremental sequence with negligible data distribution shifts, by randomly splitting ImageNet-1K (Russakovsky et al., 2015) into four independent and identically distributed (IID) data chunks, the random class incremental sequence with moderate data distribution shifts, by randomly splitting 1K classes of images into four disjoint chunks each with 250 classes, the distant class incremental sequence with severe data distribution shifts, by splitting 1K classes of data into four chunks while maximizing the semantical dissimilarity among chunks, and the domain incremental sequence with severe domain distribution shifts, by taking each domain in DomainNet (Peng et al., 2019) as a data chunk.

As for the evaluation, we consider three downstream tasks following Ericsson et al. (2021), including few-shot evaluation and linear evaluation (also named many-shot classification) on 12 image classification datasets (Kornblith et al., 2019b), and the Pascal VOC (Everingham et al., 2010) detection task. Through extensive experiments with more than 500 pre-trained models, we thoroughly investigate key roles in sequential SSL, including streaming data, downstream tasks and datasets, continual learning methods, SSL methods, and the method efficiency in terms of time and storage. We also thoroughly investigate the knowledge forgetting behavior of sequential SSL and supervised learning (SL) models and provide a comprehensive empirical analysis of the underlying reason.

To the best of our knowledge, we are among the first to explore the sequential self-supervised pre-training setting and the first to provide a thorough empirical study on self-supervised pre-training with streaming data. We summarize the takeaways as well as our contributions as: *i).* Sequential SSL models exhibit the on-par transfer learning performance as joint SSL models on streaming data with negligible or mild distribution shifts. As for streaming data with severe distribution shifts or longer sequences, i.e., the distant class incremental sequence, evident performance gaps exist between sequential SSL and joint SSL models. Such performance gaps, however, can be mitigated effectively and efficiently with unsupervised parameter regularization (Aljundi et al., 2018) and simple data replay. *ii).* Based on the above finding, we conclude that the standard joint training paradigm may be unnecessary for SSL pre-training. Instead, sequential SSL is performance-competitive but more time-efficient and storage-saving and is well worth considering as the practical practice for self-supervised pre-training with streaming data. *iii).* Compared with supervised learning (SL) models, SSL models consistently show smaller performance gaps between ST and JT. Our comprehensive investigation of learned representations demonstrates that sequential SSL models are less prone to

catastrophic forgetting than SL models. *iv).* Through the empirical analysis on the sharpness of minima in the loss landscape, we find that SSL models have wider minima than SL models, which we hypothesize is the reason for less forgetting of SSL models.

## 2 RELATED WORK

**Self-supervised learning (SSL).** SSL learns useful features by solving various pretext tasks using supervisions generated from unlabeled training data, e.g., predicting rotations (Gidaris et al., 2018), predicting cluster assignments (Caron et al., 2018), and solving instance discrimination (Wu et al., 2018; Chen et al., 2020a; He et al., 2020; Grill et al., 2020). To achieve better performance in the downstream task, recent studies of SSL have made efforts in either upstream pre-training or downstream transfer (Zhang et al., 2021). Previous works (Caron et al., 2019; He et al., 2020) have leveraged especially large datasets for pre-training, such as YFCC 100M (Thomee et al., 2016) and Instagram 1B (Mahajan et al., 2018). Some recent works (Gururangan et al., 2020; Reed et al., 2021) propose to pre-train with the downstream dataset for a better transfer. Our work still focuses on the downstream-agnostic model pre-training. However, in realistic scenarios, access to massive data is often streaming, and how to perform SSL with streaming data has not been studied before, motivating our work.

**Continual learning.** Existing studies of continual learning (CL) (Delange et al., 2021) mainly focus on supervised tasks and can be summarized into three categories, including regularization, replay, and parameter-isolation. In regularization-based CL, knowledge preserving is achieved by regularizing the parameter posterior of the new task not to deviate drastically from the prior (Aljundi et al., 2018; Kirkpatrick et al., 2017; Zenke et al., 2017). Replay-based CL methods overcome forgetting by saving samples of previous tasks in a replay buffer (Rebuffi et al., 2017; Rolnick et al., 2019; Wang et al., 2021; Yan et al., 2021b) and using them to regularize the learning of new tasks. Last, isolation-based CL methods leverage different parameters for learning each task to preserve the learned knowledge (Serra et al., 2018; Mallya & Lazebnik, 2018). Although works (Rao et al., 2019; Aljundi et al., 2019) explore continual learning for some specific unsupervised tasks, few have studied the transfer learning performance of sequential self-supervised models.

## 3 PROBLEM SETTING

In pre-training tasks, we train representation models on large-scale datasets, such as ImageNet (Russakovsky et al., 2015), and evaluate the transferability of learned representations on various downstream tasks (Chen et al., 2020a). In our empirical study, we adopt the prevailing MoCo-v2 (Chen et al., 2020c) method to pre-train SSL models with diverse streaming data.

**Types of streaming data.** In pre-training, we consider streaming data with various distribution shifts to mimic practical data collection scenarios. As shown in Figure 1, each type of streaming data consists of sequential and disjoint data chunks, while collective data cover all available data. In the instance incremental sequence, streaming data chunks are almost IID, which simulates the scenario where data are continually collected under the same condition. In this case, there is negligible distribution shift among sequential data chunks. In the random class incremental sequence, data in disjoint chunks belong to different classes, which mimics the scenario where data are collected by random keyword search on the Internet (Parisi et al., 2019). Here the distribution shift is moderate. The distant class incremental sequence is similar to the random class incremental sequence except that the semantic gaps between sequential data chunks in the distant sequence are larger, i.e., images from different data chunks are semantically dissimilar. This data sequence has severe distribution shifts between chunks. It mimics the scenario where data are crawled from websites with different subjects. In the domain incremental sequence, data chunks are collected from different domains with severe domain distribution shifts. A typical example is that large-scale autonomous driving data in Han et al. (2021) are collected in different domains, such as different weather conditions and cities, but share similar classes. The first three types of streaming data are designed with ImageNet (Russakovsky et al., 2015), while the domain incremental sequence consists of five domains in DomainNet (Peng et al., 2019). See Appendix A.1 for a detailed description.

**Model pre-training.** With these streaming data, we study both sequential training (ST) and joint training (JT) for model pre-training. As illustrated in Figure 1, in sequential training, a model is

sequentially trained with streaming data chunks, while in joint training, a model is repeatedly trained with collective data, i.e., all seen data chunks. Moreover, we compare SSL with supervised learning (SL) and mainly study the following pre-trained models: sequentially trained SSL models (SSL-ST), jointly trained SSL models (SSL-JT), sequentially trained SL models (SL-ST), and jointly trained SL models (SL-JT). See Appendix A.2 for details of the pre-training stage.

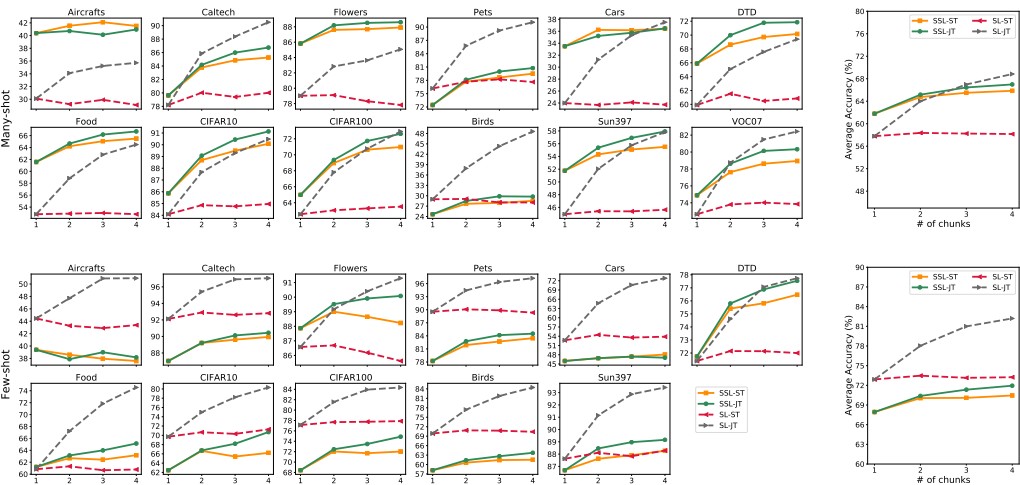

Figure 2: Linear and few-shot evaluation results of **random class incremental sequence**. On the left are the results of each dataset. On the right are averaged results across all left datasets.

**Transfer to downstream tasks.** We evaluate the transfer learning performance of pre-trained models using three typical downstream tasks: many-shot classification, few-shot classification, and object detection. Following Chen et al. (2020a), we consider 12 classification datasets for downstream evaluation. Specifically, we conduct many-shot classification on all the above 12 datasets but conduct few-shot classification on 11 datasets except VOC2007, following Ericsson et al. (2021). In both types of classification tasks, representations are fixed for evaluation. In addition, we evaluate pre-trained models on the PASCAL VOC detection dataset following He et al. (2020). See Appendix A.4 for more details of the downstream evaluation.

## 4 DISSECTION OF SEQUENTIAL SELF-SUPERVISED PRE-TRAINING

We pre-train representation models on four types of streaming data and evaluate pre-trained models on 12 downstream datasets with three downstream evaluation tasks. Note that models pre-trained with ImageNet-based streaming data are evaluated on all three downstream tasks. Models trained with the domain incremental sequence are only evaluated with few-shot classification, considering the size of DomainNet is only $1/5$ ImageNet. We report downstream evaluation results of the random class incremental sequence, the distant class incremental sequence, and the domain incremental sequence in Figure 2, Figure 3, and Figure 4, respectively. Results of the instance incremental sequence are illustrated in Appendix B.1 since the performance of SSL models on the instance incremental sequence is similar to that on the random class incremental sequence without considering the different cases for SL models. We also evaluate three types of ImageNet-based streaming data on object detection and illustrate results in Figure 10 in Appendix.

### 4.1 HOW DOES TRANSFER LEARNING PERFORMANCE VARY WITH STREAMING DATA?

We first consider streaming data with various degrees of distribution shifts, i.e., streaming data with negligible distribution shifts such as the instance incremental sequence, streaming data with moderate distribution shifts such as the random class incremental sequence, and streaming data with severe distribution shifts such as the distant class incremental sequence and the domain incremental sequence. As shown in Figures 2-4, on all types of streaming data, the performance of sequential SSL models generally increases with more streaming chunks, while sequential SL models do not

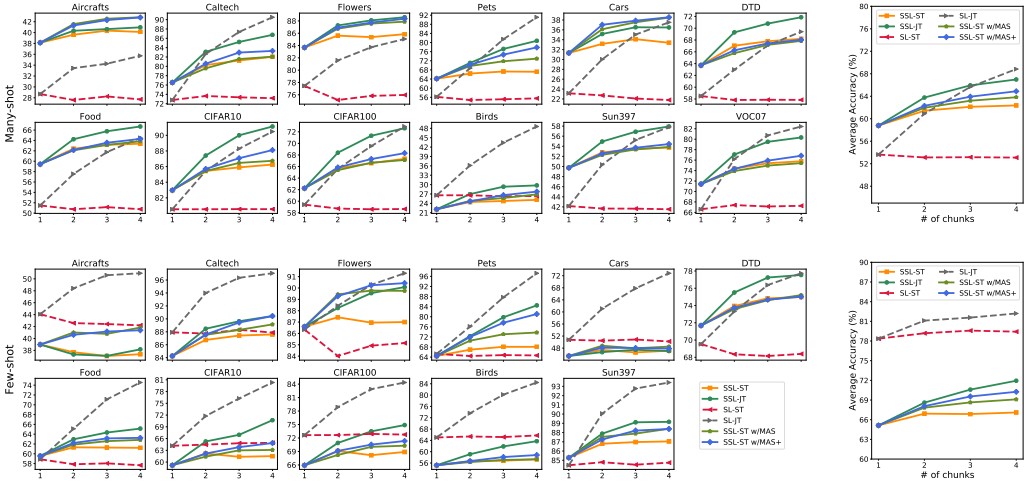

Figure 3: Linear and few-shot evaluation results of **distant class incremental sequence**. On the left are the results of each dataset. On the right are averaged results across all left datasets.

benefit from increasing data chunks. As for the performance on each type of streaming data, sequential SSL models surprisingly perform comparably to joint SSL models on streaming data with negligible and moderate distribution shifts. On streaming data with severe distribution shifts, the performance of sequential SSL models is evidently inferior to that of joint SSL models. The detection evaluation results in Figure 10 in Appendix further support this observation. In addition, we provide results of streaming data with longer chunks (8) and random distribution shifts in Appendix B.2. Similarly, we find the long sequence leads to visible but not significant gaps between ST and JT models. In contrast, on all types of streaming data, sequential SL models perform especially worse than joint SL models. The above observations denote that, unlike traditional continual learning tasks (Delange et al., 2021), although faced with possible visible performance gaps between ST models and JT models, sequential SSL is still performance-promising in pre-training tasks with streaming data.

## 4.2 ARE RESULTS CONSISTENT ACROSS DOWNSTREAM TASKS OR DATASETS?

In pre-training tasks, we pay attention to the generalization of the learned representations to new data or tasks rather than the performance on the training dataset. Taking a closer look at the results in Figures 2-4, we observe that, although joint SSL models achieve comparable performance to joint SL models in linear evaluation, joint SL models significantly outperform joint SSL models in few-shot evaluation. This observation is also demonstrated in (Tian et al., 2020; Ericsson et al., 2021). The main difference between the two evaluation protocols is that linear evaluation involves more fine-tuning than few-shot evaluation, as introduced in Appendix A.4. Therefore, the underlying reason for the observation is that supervised features are correlated with labels and more discriminative, thus easy to directly transfer to downstream datasets similar to upstream pre-training data (DomainNet or ImageNet). For example, SL models dominate most few-shot object or scene classification tasks but fail on DTD (Cimpoi et al., 2014), a texture classification dataset sharing no common classes with ImageNet or DomainNet. In contrast, self-supervised features are more generalized and comprehensive, thus requiring more fine-tuning for desirable downstream transfer. In addition, on some downstream datasets, we have seemingly abnormal observations that ST models may outperform JT models and the model performance may drop with the increase of chunk number. These phenomena are due to the so-called "negative transfer" (Wang et al., 2019), which is also discussed in other model pre-training studies (Newell & Deng, 2020; Gururangan et al., 2020). That is, pre-training with more data chunks does not necessarily benefit a specific downstream dataset if the added training data are irrelevant to the downstream dataset. See Appendix B.3 for a concrete example of "negative transfer" on Oxford-IIIT Pets (Parkhi et al., 2012) in pre-training with streaming data. It is observed that sequential SSL models suffer less "negative transfer" than SL models and continual learning methods largely prevent "negative transfer".

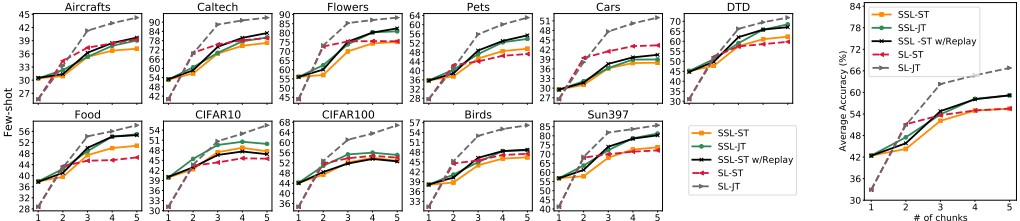

Figure 4: Few-shot evaluation results of **domain incremental sequence**. On the left are the results of each dataset. On the right are averaged results across all left datasets.

## 4.3 DO CONTINUAL LEARNING METHODS HELP SEQUENTIAL SSL?

As observed in Section 4.1, there still exist obvious performance gaps between sequential SSL models and joint SSL models, on streaming data with severe distribution shifts such as the distant class incremental sequence and the domain incremental sequence. To this end, we study whether continual learning methods can help mitigate such gaps. Specifically, we investigate two classic methods in continual learning, i.e., data replay and MAS (Aljundi et al., 2018), which are effective to defy knowledge forgetting in supervised classification tasks (Delange et al., 2021). When using data replay, we randomly reserve 10% data from each seen data chunk and add them to the current data chunk for model pre-training. We also consider the combination of MAS and data replay, which is referred to as MAS+ in the experiments. We denote SSL models trained with data replay as SSL-ST w/Replay, SSL models trained with MAS as SSL-ST w/MAS, and SSL models trained with both methods as SSL-ST w/MAS+. We report downstream evaluation results of the distant class incremental sequence in Figure 3 and results of the domain incremental sequence in Figure 4. As shown in Figure 4, data replay can totally eliminate the performance gaps between sequential SSL models and joint SSL models on the domain incremental sequence. Results in Figure 3 also validate the effectiveness of both continual learning methods in improving the transfer learning performance of sequential SSL models trained with streaming data with severe distribution shifts. In short, we find methods devised for supervised continual tasks are especially promising to make sequential SSL models perform comparably to joint SSL models on challenging streaming data. See Appendix A.3 for implementations of MAS and data replay in sequential SSL.

## 4.4 HOW ABOUT SSL METHODS OTHER THAN MOCO?

For simplicity, we choose MoCo-v2 (Chen et al., 2020c) in experiments and demonstrate that sequential SSL is performance-promising. To verify whether it also holds for other SSL methods, we train BYOL (Grill et al., 2020) models on the challenging distant class incremental sequence, both sequentially and jointly. Results of BYOL are shown in Figure 13 in Appendix B.4. Similar to MoCo-v2, there still exist visible performance gaps between sequential SSL models and corresponding joint SSL models. In contrast, SSL models exhibit much smaller performance gaps than SL models, which further validates the potential of sequential SSL in pre-training tasks.

## 4.5 ANALYSIS OF METHOD EFFICIENCY

We then discuss the time and memory consumption of different training methods of SSL, including sequential training (SSL-ST), ST with data replay (SSL-ST w/Replay), ST with MAS (SSL-ST w/MAS), ST with MAS and data replay (SSL-ST w/MAS+), and joint training (SSL-JT). As shown in Table 1, JT is very time-consuming especially when the data amount is large, while ST is able to save a large amount of time under sequential training scenarios. To be specific, ST is about 2x faster than JT when there are 2 chunks of data, and is about 4x faster when the number of chunks is 4. Moreover, when we use MAS and data replay to improve the performance of ST, the time consumption of SSL increases a little but is still significantly faster than JT. As for storage consumption, we can observe a similar phenomenon as shown in Table 1. In summary, sequential SSL is much more time-efficient and storage-saving than JT, especially when the data amount is large or grows quickly. Such a result indicates that sequential SSL is a more favorable choice for real-world pre-training applications, where data come in sequentially and grow daily.

Table 1: Resource efficiency of considered SSL pre-training methods. We take the distant class incremental sequence as an example and report the training time (h) and required storage (GB) of the model pre-trained with each data chunk. Note that all the following statistics are recorded under the same hardware environment. The lower value means better efficiency.

| Time (Storage) / Chunk | 2 | 3 | 4 |
|---|---|---|---|
| SSL-ST | **16.5 (35)** | **16.5 (35)** | **16.6 (35)** |
| SSL-ST W/Replay | 17.0 (35) | 18.5 (42) | 20.0 (46) |
| SSL-ST w/MAS | 18.2 (35) | 18.1 (35) | 18.1 (35) |
| SSL-ST w/MAS+ | 22.4 (39) | 24.4 (42) | 26.4 (46) |
| SSL-JT | 31.1 (70) | 46.5 (105) | 66.6 (140) |

Table 2: The comparison of pre-training methods in terms of the transfer performance gap between ST and JT models. We report the averaged accuracy gaps of linear evaluation across 12 downstream datasets. The lower, the better.

| Accuracy gap (%) / Chunk | 2 | 3 | 4 |
|---|---|---|---|
| SL-ST (Instance) | 2.26 | 3.27 | 4.83 |
| SSL-ST (Instance) | **0.41** | **1.02** | **1.04** |
| SL-ST (Random) | 5.63 | 8.73 | 10.68 |
| SSL-ST (Random) | **0.42** | **0.94** | **1.13** |
| SL-ST (Distant) | 7.77 | 12.50 | 15.75 |
| SSL-ST (Distant) | 2.34 | 3.81 | 4.62 |
| SSL-ST w/MAS (Distant) | 1.82 | 2.73 | 3.17 |
| SSL-ST w/MAS+ (Distant) | **1.47** | **2.01** | **2.10** |

**Summary.** We show the averaged accuracy gaps between ST models and the corresponding JT models under linear evaluation in Table 2, for both SSL and supervised learning (SL). On streaming data with negligible distribution shifts, SL exhibits evident accuracy gaps while SSL has negligible gaps. On streaming data with moderate distribution shifts, SL exhibits larger accuracy gaps while SSL still keeps the negligible gaps. On streaming data with severe distribution shifts, SL shows much larger accuracy gaps, while SSL shows mild accuracy gaps. But such accuracy gaps of SSL can be effectively mitigated with simple continual learning methods. To sum up, SSL exhibits significantly smaller performance gaps between ST models and JT models than SL. The above difference between SL and SSL models motivates us to further investigate the forgetting property in Section 5.

## 5 SELF-SUPERVISED MODELS FORGET LESS THAN SUPERVISED MODELS

In this section we first analyze the knowledge forgetting of previous tasks from two perspectives. In Section 5.1, we evaluate the transfer ability of both SL and SSL representations via the standard backward and forward transfer analysis in continual learning (Lopez-Paz & Ranzato, 2017). In Section 5.2, we adopt the CKA similarity (Kornblith et al., 2019a) and image reconstruction from features (Zhao et al., 2020) to directly analyze the representation. Last but not least, in Section 5.3, we provide our empirically justified hypothesis for why SSL models forget less than SL models.

### 5.1 BACKWARD AND FORWARD TRANSFER ANALYSIS OF SEQUENTIAL LEARNING

Following (Lopez-Paz & Ranzato, 2017), we adopt the backward and forward transfer to assess the knowledge transfer in sequential learning. Backward transfer refers to the improvement of performance on previously learned chunks when learning new chunks, where large negative transfer is also known as catastrophic forgetting. Forward transfer measures the improvement in performance on the novel chunk with the accumulation of knowledge from previous chunks. For supervised continual learning, the performance

Table 3: Backward and forward transfer analysis of sequential learning.

| Data | Method | BWT(%) | | FWT(%) | |
|---|---|---|---|---|---|
| | | Top-1 | Top-5 | Top-1 | Top-5 |
| Instance | SL | -9.45 | -5.46 | **8.64** | 2.81 |
| | SSL | **3.61** | **3.60** | 7.55 | **8.63** |
| Random | SL | -20.63 | -7.03 | -0.34 | 0.01 |
| | SSL | **-5.17** | **-1.36** | **11.05** | **4.52** |
| Distant | SL | -40.43 | -28.66 | 4.90 | 0.47 |
| | SSL | **-13.24** | **-11.06** | **11.01** | **3.66** |

is defined as the accuracy on the associated test set, which is meaningful due to the consistency of the training and test sets. Similarly, we conduct the whole analysis based on the performance on the pre-training data chunks, instead of performing the evaluation on downstream datasets. Specifically, for the ease of comparison between SL and SSL, we measure the performance by KNN classification accuracy on the representations of pre-training chunks, where the labels are provided just for evaluation, similar to Wu et al. (2018). Concretely, the backward transfer BWT $= \frac{1}{T-1} \sum_{i=2}^{T} \frac{1}{i} \sum_{j=1}^{i} A^i_{\mathcal{Y}_j} - A^j_{\mathcal{Y}_j}$ and forward transfer FWT $= \frac{1}{T-1} \sum_{i=2}^{T} A^i_{\mathcal{Y}_i} - \tilde{A}_{\mathcal{Y}_j}$ metrics used in Yan et al. (2021a) where $T$ is the sequence length, $A^i_{\mathcal{Y}_j}$ refers to the accuracy on the chunk $j$ using model learned at step $i$ where the label space includes all observed classes up to chunk $j$, and $\tilde{A}_{\mathcal{Y}_j}$ means the accuracy with the model learned from scratch. The results are shown in Table 3, and

the implementation details are included in the Appendix C.2. We can obtain the following observations about forgetting: *i).* Learning method: SSL itself is less prone to catastrophic forgetting than SL, especially that SSL achieves positive backward transfer on the instance incremental sequence. It illustrates that SSL is more suitable for streaming data. *ii).* Types of streaming data: The model suffers progressively severe forgetting when the distribution shift increases for both SSL and SL cases. *iii).* Example forgetting: It is observed that forgetting is less severe in top-5 classification than top-1 classification, which indicates that the knowledge is not fully forgotten.

## 5.2 REPRESENTATION MEMORIZATION ANALYSIS OF SEQUENTIAL LEARNING

**How do features forget in sequential training?** We study how learned features forget in sequential training via Centered Kernel Alignment (CKA) (Kornblith et al., 2019a). CKA is used to measure the similarity between two representations of the same given samples. See Appendix C.3 for details of the CKA similarity. Specifically, we randomly sample 50,000 images from the first data chunk on each type of streaming data. We use these samples and sequentially trained models for CKA similarity analysis. We report the CKA similarity values on three types of ImageNet-based streaming data in Figure 5. Each value is obtained by first extracting features of samples with two different models and then computing the CKA feature similarity value between the two features. On all streaming data, we have three consistent observations about the CKA similarity between sequential models: *i).* SSL models all exhibit higher features similarity to the initial model, compared with SL models. *ii).* In general, SSL models show higher features similarity between two sequential models in sequential training, compared with SL models. *iii).* Features similarity between two sequential models decrease on streaming data with more severe distribution shifts, for both SSL and SSL.

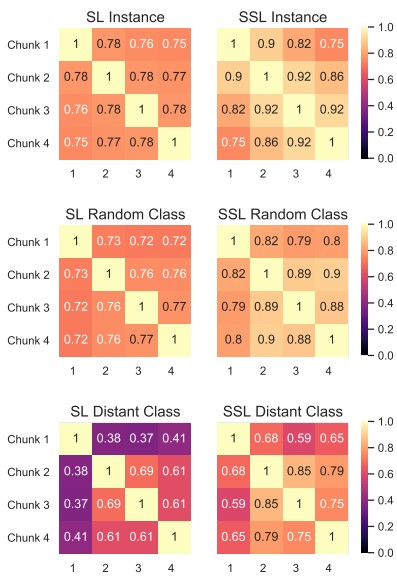

Figure 5: CKA scores between sequentially trained models.

These observations suggest that features of SSL models forget less and evolve more slowly than those of SL models in sequential training.

**Image reconstruction by feature inversion for sequential models.** Similar to Zhao et al. (2020), in Figure 6, we visualize images reconstructed from both SL-ST and SSL-ST features using deep image prior (DIP) (Ulyanov et al., 2018). To be specific, we choose four images in the first data chunk of the challenging distant class incremental sequence and visualize features of four sequentially learned models for both SSL and SL, respectively. As shown in Figure 6, in sequential training, features of SSL models can always perfectly reconstruct the main information in original images. In contrast, features of SL models lose more detailed information with more sequential data chunks, which indicates SSL is much better at countering the knowledge forgetting in sequential training. Recalling the evolving CKA similarity shown in Figure 5, the perfect reconstruction results of sequential SSL models do not mean SSL models stop learning in sequential training. Instead, it indicates that SSL does well in learning new knowledge while keeping previous knowledge.

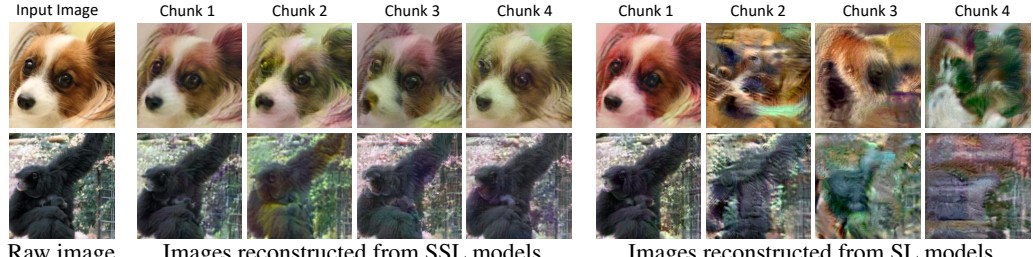

Figure 6: Images reconstruction by inversing features from both SSL and SL models in sequential pre-training.

Table 4: Comparisons of the sharpness of minima between SL and SSL models. Lower is better.

| | Instance | | Random Class | | Distant Class | |
| | $\epsilon = 0.1$ | $\epsilon = 0.3$ | $\epsilon = 0.1$ | $\epsilon = 0.3$ | $\epsilon = 0.1$ | $\epsilon = 0.3$ |
|---|---|---|---|---|---|---|
| SL | 0.47 | 0.94 | 0.21 | 0.94 | 0.19 | 0.94 |
| SSL | **0.14** | **0.68** | **0.08** | **0.66** | **0.12** | **0.71** |

### 5.3 HYPOTHESIS FOR DIFFERENT FORGETTING BEHAVIORS BETWEEN SL AND SSL

In this subsection, we dig into the different forgetting behavior between SL and SSL by analyzing the sharpness of the minima in the loss landscape. Flat minima in the loss landscape are the minima in which the change of losses is slow in its neighborhood. Note that the models having flat minima in the loss landscape tend to exhibit an impressive generalization ability (Keskar et al., 2016). When starting with flat minima, we expect that learning new chunks will have a minor effect on the performance of existing chunks, as escaping the wide basin is difficult. Therefore, we hypothesize that SSL encourages the model to seek out flatter minima, which increases SSL's resistance to catastrophic forgetting. To verify this hypothesis, we conduct experiments to compare the sharpness of minima between SL and SSL models, where we apply a widely-used sharpness metric (Keskar et al., 2016; Wen et al., 2018). Concretely, we first define the neighborhood $\mathcal{C}_\epsilon$ of mimina as:

$$\mathcal{C}_\epsilon = \{z \in \mathbb{R}^n : -\epsilon||\boldsymbol{\theta}||_2 \leq ||z||_2 \leq \epsilon||\boldsymbol{\theta}||_2\}, \tag{1}$$

where $n$ denotes the number of parameters and $\boldsymbol{\theta}$ refers to the model parameter after training. Because SSL and SL models are trained with different loss objectives, such as cross-entropy loss and the contrastive loss, we cannot directly analyze the sharpness with either objective. Considering that we aim for a representation model, we propose to directly adopt the KNN classifier to evaluate representations of both SL and SSL models. The KNN classification loss can be a discrete proxy of loss functions. Then, the sharpness of loss minima $\Phi_{\boldsymbol{\theta},f}$ is defined as follows:

$$\Phi_{\boldsymbol{\theta},f}(\epsilon) = \max_{\boldsymbol{\theta}' \in \mathcal{C}_\epsilon} g(\boldsymbol{\theta}', \boldsymbol{\theta}) = \max_{\boldsymbol{\theta}' \in \mathcal{C}_\epsilon} \frac{f(\boldsymbol{\theta}) - f(\boldsymbol{\theta}')}{f(\boldsymbol{\theta})}, \tag{2}$$

where $g(\boldsymbol{\theta}', \boldsymbol{\theta})$ means the relative loss change from minima $\boldsymbol{\theta}$ to the parameter $\boldsymbol{\theta}'$, and the loss function $f(\boldsymbol{\theta})$ is the negative KNN classification accuracy with model parameter $\boldsymbol{\theta}$.

As shown in Table 4, SSL indeed discovers flatter minima compared to SL, which verifies our hypothesis and provides an explanation for why SSL suffers less forgetting than SL. More implementation details are in Appendix C.4. Moreover, we also conduct the visualization of relative loss change $g$ over a linear path like (Mirzadeh et al., 2020) in Appendix C.4. The loss change of SSL is slower than that of SL along the linear interpolation path, demonstrating the flatter minima of SSL.

## 6 DISCUSSIONS

This paper has conducted the first thorough empirical evaluation to investigate how well self-supervised learning (SSL) performs with various streaming data types and diverse downstream tasks. Our experimental results and the empirical analysis conclude the three main findings: *i).* Joint training is unnecessary for SSL with streaming data. Instead, sequential training with suitable continual learning strategies is performance-competitive yet more efficient, well worth considering as a good alternative. *ii).* Sequential self-supervised pre-training shows a better capability of overcoming catastrophic forgetting than sequential supervised pre-training. *iii).* We hypothesize that SSL models have flatter minima than SL models in the loss landscape, which seems reasonable for the different forgetting behaviors between SL and SSL models. Moreover, We demonstrate this hypothesis by a thorough empirical analysis of the sharpness of minima.

As for future directions, we first call for more attention to sequential self-supervised learning for understanding its underlying theories of knowledge forgetting and devising better approaches. Also, we recommend considering sequential self-supervised training as a more efficient representation learning paradigm for real-world applications.

ACKNOWLEDGMENTS

This work is supported by NUS ARTIC Project (Project Reference: ECT-RP2) and NRF Centre for Advanced Robotics Technology Innovation (CARTIN). Dr. Lanqing Hong and Prof. Xinchao Wang are corresponding authors. We thank Mr. Jiawei Du for his help with the sharpness metric and Mr. Yujun Shi for his discussion on the feature uniformity. We also thank the anonymous reviewers for their valuable comments.

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

APPENDIX

## A EXPERIMENTAL SETUPS

### A.1 TYPES OF STREAMING DATA

We consider four kinds of streaming data for the study, i.e., the instance incremental sequence, the random class incremental sequence, the distant class incremental sequence, and the domain incremental sequence. To exclude the effect of the number of images, we make sure that data chunks in the same sequence have almost the same data amount. Here we provide more details about these data sequences.

**Instance incremental sequence.** For the instance incremental sequence, we split the ImageNet (Russakovsky et al., 2015) training data that consists of 1.28 million images with 1,000 classes into four even chunks. We ensure that each data chunk includes the same 1,000 classes with the same number of images for each class, which means these data chunks are independent and identically distributed (IID).

**Random class incremental sequence.** For the random class incremental sequence, we randomly split the 1,000 classes of ImageNet into four parts where each part has 250 classes. Since each class of ImageNet has around 1,000 images, we can directly obtain four data chunks with almost the same amount of images.

**Distant class incremental sequence.** To explore the data sequence with severe distribution shifts among data chunks, we consider the distant class incremental sequence. Following (Yosinski et al., 2014), rather than randomly splitting the 1,000 classes, we leverage the WordNet Tree (Miller, 1998) to obtain four even data chunks sharing the minimal semantic overlapping. We first build a 1000*1000 adjacent matrix among the 1,000 classes by setting the value of similar classes as 1 and the value of dissimilar classes as 0. To be specific, we take classes sharing the common parent node beneath the ninth depth in the WordNet Tree as similar classes and vice versa. Using the semantic similarity described in the adjacent matrix, we then split the 1,000 classes into independent connected components as shown in Figure 7. Finally, we merge these imbalanced components into four almost even data chunks. Concretely, the first chunk 'A' includes 250 classes of 318,459 images. The second data chunk 'B' includes 250 classes of 321,488 images. The third data chunk 'C' includes 251 classes of 321,533 images. The fourth data chunk 'D' includes 249 classes of 319,687 images.

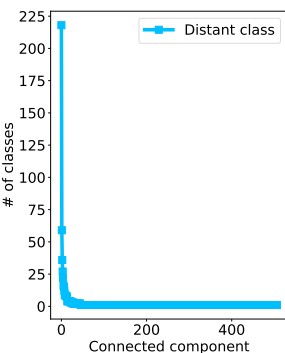

Figure 7: The number of classes for each connected component from the adjacent matrix of 1,000 classes in ImageNet.

**Domain incremental sequence.** As for the domain incremental sequence, we consider a multi-domain dataset called DomainNet (Peng et al., 2019). In our work, we adopt a domain incremental data sequence made of four distant domains including 'sketch', 'real', 'painting' 'quickdraw', and 'clipart'. There exist severe domain distribution shifts among data in these five domains. Specifically, data in the domain 'quickdraw' mostly contain only lines without visual textures. As a result, images from 'quickdraw' are less informative and more visually distinct, compared with images from those four domains, as shown in Figure 8. For each domain, we randomly select 48,129 images as a data chunk, except for 'quickdraw' where we select 47,687 images.

### A.2 DETAILS OF PRE-TRAINING

**MoCo-v2.** For the illustration purpose, we adopt a prevailing self-supervised learning (SSL) method, MoCo-v2 (Chen et al., 2020c), to investigate the performance of SSL with streaming data. MoCo-v2 uses a Siamese network consisting of two encoders. These two encoders are designed for query images and key images, respectively, and share the same architecture where an MLP projection head $f_w$ is on top of a backbone network $f_\theta$. Only the query encoder is updated by the gradients

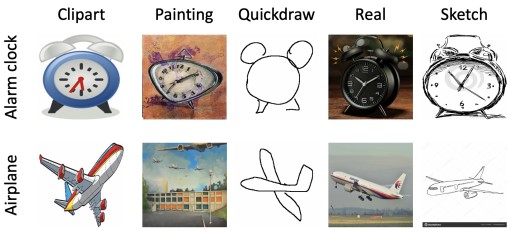

Figure 8: Example images in the five domains of DomainNet.

backpropagation while the key encoder is updated by the moving average with a momentum. MoCo-v2 maintains an additional dictionary as a queue of features for contrastive learning. Specifically, features in the dictionary are progressively updated. The current mini-chunk features from the key encoder are enqueued and the same number of oldest features are dequeued. MoCo-v2 uses In-foNCE (Oord et al., 2018), a variant of contrastive loss (CL), to maximize the similarity of features from positive pairs and minimize the similarity of features from negative pairs. The contrastive loss is formalized as below.

$$\mathcal{L}_{cl} = -\frac{1}{N} \sum_{i=1}^{N} \log \frac{e^{(z_i^\top z_i^+ / \tau)}}{e^{(z_i^\top z_i^+ / \tau)} + \sum_{z_i^- \in Z^-} e^{(z_i^\top z_i^- / \tau)}}, \tag{3}$$

where $N$ is the number of samples, $z_i$ is the L2-normalized projected feature from the query encoder, $z_i^+$ is the L2-normalized projected feature of the same input image from the key encoder, $Z^-$ are the negative history features stored in the dictionary and $\tau$ is the temperature.

**Pre-training.** For self-supervised pre-training, we follow the protocol of MoCo-v2 (Chen et al., 2020c), i.e., using the standard ResNet50 backbone (He et al., 2016). The implementation is based on OpenSelfSup[1]. For both joint training and sequential training, the number of training epochs is 200 for each model training, where the convergence of loss is observed, as shown in Figure 14. As for the queue in MoCo-v2 training, we adopt the same as the original MoCo-v2, i.e., the queue size is 65,536. For data chunks fewer than 65,536 like DomainNet chunks, we store all samples of the current chunk for contrastive learning. In sequential training, the queue is refreshed before training on a new chunk. We never preserve previous data in the queue, to ensure the model training is sequentially conducted on disjoint chunks. For the possibly used data replay strategy, another special replay buffer is used to preserve a small fraction of data from each previous chunk, like 10% in our experiments. For the instance incremental sequence, we consider one random sequence as the data are randomly divided. While for the random class incremental sequence, distant class incremental sequence, and domain incremental sequence, we experiment with different sequences of data chunks. In particular, considered sequences are obtained through right circular shift operations. For example, if the data sequence length is 4, after splitting all the data into four chunks A, B, C, and D, four sequences, namely A-B-C-D, B-C-D-A, C-D-A-B, and D-A-B-C are used for the sequential pre-training a representation model. The results from different sequences are averaged to obtain the final performance. For comparison, supervised pre-training is also implemented using OpenSelf-Sup following the recommended training protocol of ImageNet. For supervised pre-training, the classifier layer is reset at a new data chunk.

## A.3   DETAILS OF CONTINUAL LEARNING METHODS

Continual learning is assumed to suffer from catastrophic forgetting of previously learned knowledge in supervised learning (Goodfellow et al., 2013; Kirkpatrick et al., 2017; McCloskey & Cohen, 1989), leading to significant performance degradation of previous tasks. Here we introduce the continual learning techniques we adopt, including data replay (Rolnick et al., 2019; Rebuffi et al., 2017) and regularization-based method, e.g., Memory Aware Synapses (MAS) (Aljundi et al., 2018).

**Data replay.** Data relay is a simple yet effective method for alleviating the catastrophic forgetting problem during the continual learning process. Specifically, we need to maintain a replay buffer

---

[1]https://github.com/open-mmlab/OpenSelfSup

and store a selected subset of samples from each learned task in the buffer. Then we just retrain on samples in the replay buffer to revisit old tasks while training the model for a new task.

To perform data replay in the sequential training process, we maintain a replay buffer consisting of images sampled from previous data chunks. After finishing the sequential training with each data chunk, we randomly select 10% data of this chunk and store sampled data in the replay buffer. For the sequential training with the current data chunk, we directly mix current data with data in the replay buffer for self-supervised pre-training e.g. MoCo-v2 training using Eq. (3).

**MAS.** Regularization-based methods aim to mitigate the catastrophic forgetting by consolidating previous knowledge with an added regularization term in the loss function. One typical unsupervised regularization-based method is MAS (Aljundi et al., 2018). Specifically, MAS proposes to compute gradients the squared L2-norm of the encoder output $f_\theta$ as the importance weights of parameters.

$$\Omega_{ij} = \frac{1}{N} \sum_{k=1}^{N} \frac{\partial [\|f_\theta(x)\|_2^2]}{\partial \theta_{ij}}. \tag{4}$$

With the parameter regularization term added, the resulting loss function with the coefficient $\lambda$ is shown as below.

$$\mathcal{L}(\theta) = \mathcal{L}_{cl}(\theta) + \lambda \sum_{i,j} \Omega_{ij}(\theta_{ij} - \theta_{ij}^*)^2. \tag{5}$$

In MoCo-v2, the query encoder is considered as the representation network, we thus only impose the MAS regularization term on parameters of the query encoder. Specifically, the regularization coefficient $\lambda$ is fixed to be 100. Following the prevailing use of MAS regularization (Aljundi et al., 2018; 2019), we update the MAS importance weights $\Omega_{ij}$ to cover information of each data chunk in the sequential training process. To be specific, after finishing the sequential training with each data chunk, we leverage both the current data chunk and data in the replay buffer to estimate importance weights for the trained model using Eq. (4). Then we update the stored sequential importance weights by a cumulative moving average of current and previous estimated importance weights, following (Aljundi et al., 2019). As for the model training on the current data chunk, we apply the parameter regularization using previous importance weights and optimize the model using Eq. (5).

To sum up, besides the sequentially trained model, both above methods require extra storage for sequential self-supervised pre-training. For the 10% data replay method, we need to only keep 10% data of each previous data chunk for sequential training. For the MAS regularization method, we only require to save a set of importance weights for the model and then update the importance weights sequentially.

### A.4 DETAILS OF DOWNSTREAM TASKS

We evaluate the transfer performance of the pre-trained models using three different downstream tasks. Following (Chen et al., 2020a), we consider 12 diverse image classification datasets including Food-101 (Bossard et al., 2014), CIFAR10 (Krizhevsky et al., 2009), CIFAR100 (Krizhevsky et al., 2009), Birdsnap (Berg et al., 2014), SUN397 (Xiao et al., 2010), Standard Cars (Krause et al., 2013), FGVC Aircraft (Maji et al., 2013), VOC2007 (Everingham et al., 2010), DTD (Cimpoi et al., 2014), Oxford-IIIT Pets (Parkhi et al., 2012), Caltech-101 (Fei-Fei et al., 2004) and Oxford 102 Flowers (Nilsback & Zisserman, 2008). On these datasets, we evaluate the pre-trained models via the many-shot classification and the few-shot classification (except VOC2007). Both classification protocols are the same as (Ericsson et al., 2021). In addition, we evaluate the pre-trained models on the PASCAL VOC detection task, following the same transfer protocol of MoCo (He et al., 2020). The training data of detection come from VOC2007 and VOC2012, and the test data come from VOC2007.

**Many-shot classification.** Many-shot classification is a widely used evaluation protocol (Chen et al., 2020b; He et al., 2020). To evaluate the pre-trained representations, a linear classifier is directly added to the pre-trained feature encoder. During the downstream task evaluation, only the added linear classifier is fine-tuned using a substantial amount of downstream labeled data while the feature encoder is frozen. In this way, the downstream transfer performance can directly reflect the generalization ability of the pre-trained representation models.

**Few-shot classification.** Few-shot classification reflects how well the pre-trained models perform on downstream tasks in the few-shot learning regime. Specifically, we consider 5-way 5-shot few-shot tasks on 11 downstream classification datasets, following the few-shot setting in (Ericsson et al., 2021). Concretely, the pre-trained model is fixed for extracting representations. In contrast to many-shot evaluation, in few-shot evaluation, only a few downstream labeled data are provided to obtain prototypes for different categories and then the classification is based on the nearest prototype.

**Detection.** To further evaluate the transferability of the pre-trained models on more downstream scenarios, we consider object detection as a downstream task, where the fine-grained spatial location information is more important, compared with classification tasks. To be specific, we follow the settings in (He et al., 2020), i.e., adopting the Faster-RCNN (Ren et al., 2015) with a backbone of R50-dilated-C5 and fine-tuning all layers including the pre-trained representation network.

We perform no hyper-parameter tuning for few-shot evaluation and detection evaluation. As for the linear evaluation protocol, we adopt the logistic regression and only tune the weight decay value. The inversed weight decay values for all downstream classification datasets are given in Table 5.

Table 5: The inverse of regularization strength (weight decay value) used in many-shot logistic regression evaluation on 12 different downstream classification datasets. SSL models: self-supervised models. SL models: supervised models.

| Dataset | SSL Models | SL Models |
|---|---|---|
| Aircraft | 5623.413277133687 | 9.99999985098839 |
| Caltech-101 | 316227.7712565657 | 0.3162277621819913 |
| Flowers | 31622.77530666721 | 999.999952502551 |
| Pets | 999.999952502551 | 562.3413185099295 |
| Cars | 5623.413277133687 | 17.782794106882072 |
| DTD | 1778.2794843157246 | 0.0177827946252197 |
| Food | 177827.94843157247 | 0.0562341298247638 |
| CIFAR10 | 316227.7712565657 | 0.0562341298247638 |
| CIFAR100 | 100.00000223517424 | 0.0562341298247638 |
| Birdsnap | 1778.27948431572 | 0.1 |
| SUN397 | 100.00000223517424 | 0.0177827946252197 |
| VOC2007 | 9.99999985098839 | 0.005623413223739 |

# B   MORE EXPERIMENTAL RESULTS

## B.1   RESULTS OF INSTANCE INCREMENTAL SEQUENCE

Transfer learning results of self-supervised pre-training with the instance incremental sequence are evaluated on all three downstream tasks. For results of both many-shot classification and few-shot classification in Figure 9, we find sequential SSL performs comparably with joint SSL on all downstream datasets, with the average performance gap between sequential training and joint training less than 1%, while there exists evident gaps, more than 4%, between sequential supervised learning and joint supervised learning.

## B.2   RESULTS OF LONGER SEQUENCES

We also conduct experiments on a more realistic data sequence with longer chunks and random distribution shifts. Concretely, except the 1000 classes in ILSVRC 2012, we also randomly sample 1000 classes from the remaining classes in ImageNet-21K (Ridnik et al., 2021) to build ImageNet-2K dataset with 2.29M images. Concretely, we randomly split all samples in the ImageNet-2K dataset into 8 chunks with 0.57M images per chunk. Figure 11 summarizes the results on 8-chunk sequence. Besides the similar observations in Section 4.2, we have the following observations for the longer sequence: *i).* With more sequential chunks, the performance gaps between ST and JT models become larger for both SL and SSL. *ii).* The performance gaps of SSL are visible but not significant, compared with those of SL. On average, at all steps in sequential training, the gaps of SSL are much smaller than those of SL. *iii).* The simple data replay with 10% previous data is effective in mitigating the performance gaps of SSL for the longer sequence. To make the simple data

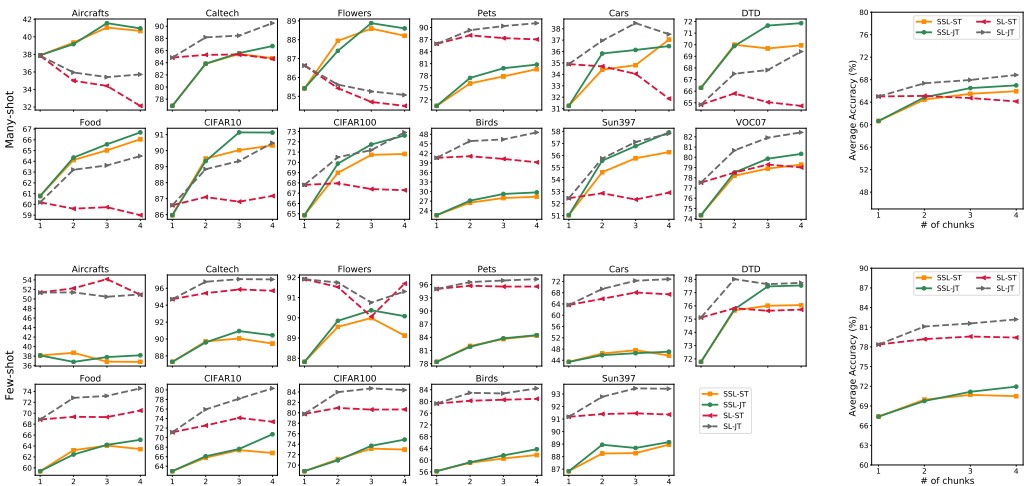

Figure 9: Linear and few-shot evaluation results of **instance incremental sequence**. on the left are the results of each dataset. On the right are averaged results across all left datasets.

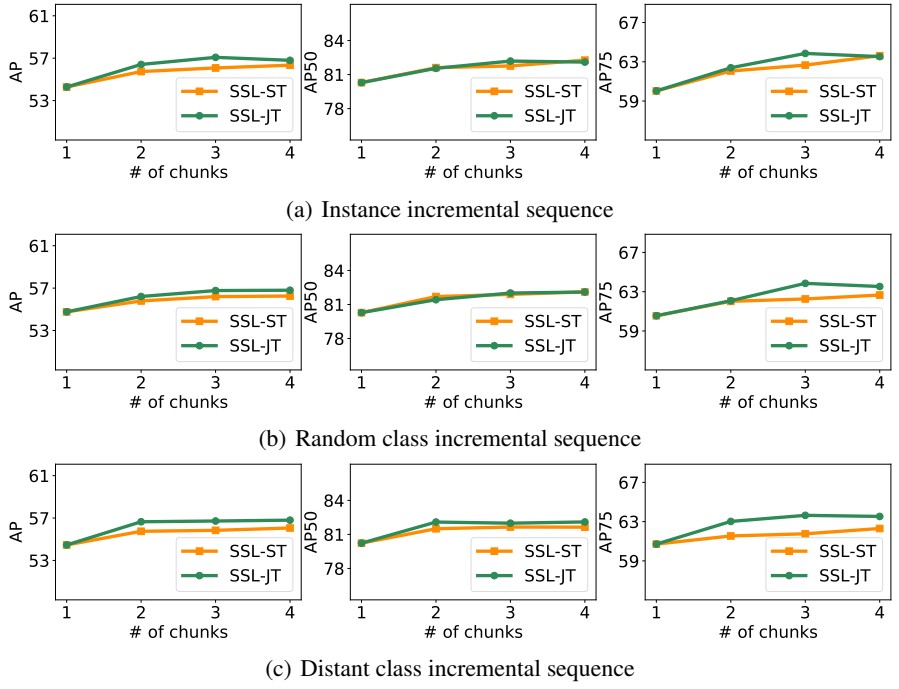

(a) Instance incremental sequence

(b) Random class incremental sequence

(c) Distant class incremental sequence

Figure 10: Object detection evaluation results of three types of ImageNet-based streaming data.

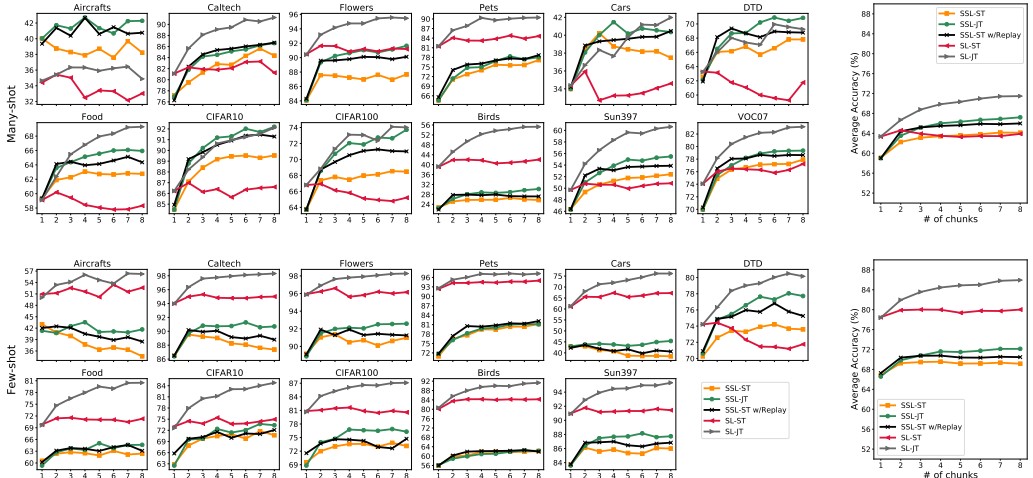

Figure 11: Linear and few-shot evaluation results of **8-chunk ImageNet-2K sequence**. on the left are the results of each dataset. On the right are averaged results across all left datasets.

replay sustainable for real streaming data with much longer chunks, we think setting an upper bound on the size of the replay buffer, such as the number of images, is feasible for realistic scenarios. To be specific, we can determine the upper bound of the replay buffer according to the physical storage limitation. Before the replay buffer approaches the upper bound, the tradeoff is only between the accuracy and training time efficiency, i.e., storing more data (larger ratio) in the replay buffer results in better accuracy but longer training time. When the number of data in the replay buffer exceeds the upper bound, we can discard some old samples to save space for data sampled from new chunks (Rebuffi et al., 2017). Generally, we believe sequential SSL with longer sequence is promising and will benefit from continual learning methods (Delange et al., 2021).

## B.3 RESULTS OF NEGATIVE TRANSFER

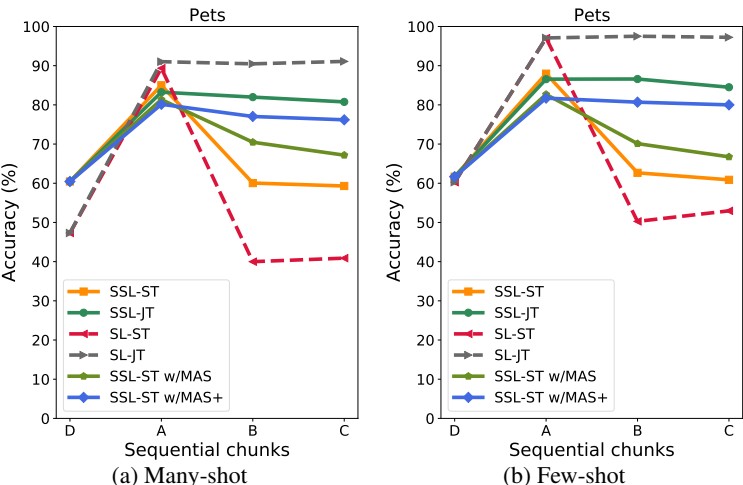

(a) Many-shot       (b) Few-shot

Figure 12: The averaged evaluation results on Pets of models pre-trained with the distant class incremental sequence D-A-B-C.

We observe severe "negative transfer" when pre-training models with distant class incremental sequence and evaluating them on Oxford-IIIT Pets (Parkhi et al., 2012). Specifically, as shown in Figure 12, SSL-ST slightly outperforms SSL-JT at the second chunk, i.e., chunk A. After chunk A,

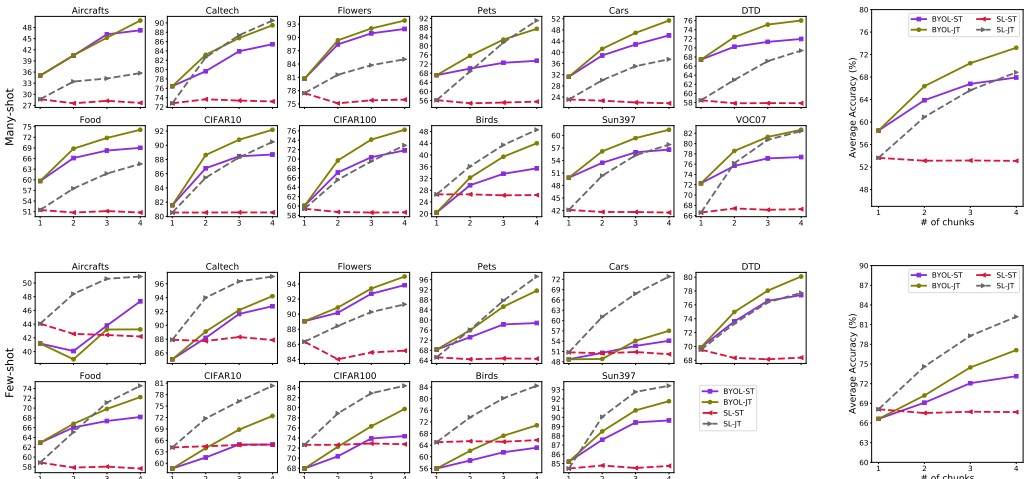

Figure 13: Linear and few-shot evaluation results of **distant incremental sequence** for BYOL. on the left are the results of each dataset. On the right are averaged results across all left datasets.

increasing more chunks for SSL-JT models, i.e., chunk B and chunk C, instead leads to significant decrease in performance. We make careful comparison between images from chunk A and images in Pets. We find that Oxford-IIIT Pets includes 37 categories of pets like dogs and cats. As mentioned in Appendix A.1, Chunk A includes many relevant classes like 'Maltese dog', 'Old English sheepdog', 'Shetland sheepdog', 'Greater Swiss Mountain dog', 'Bernese mountain dog', 'French bulldog', 'Eskimo dog', 'African hunting dog', 'tabby', 'tiger cat', 'Persian cat', 'Siamese cat', 'Egyptian cat', and 'Madagascar cat'. The other three data chunks, i.e., chunk D, B, and C, do not have these classes. Such observations are consistent with the widely accepted belief that "transferring knowledge from the dissimilar source can have a negative impact on the target learner" (Wang et al., 2019). Note that SL models suffer severer 'negative transfer' than SSL models and continual learning methods can help SSL models significantly reduce the negative transfer, achieving on-par performance to SSL-JT models.

## B.4   RESULTS OF BYOL

To evaluate whether sequential training performs well for other SSL methods, we conduct the challenging distant class incremental sequence experiments with BYOL (Grill et al., 2020). The results of BYOL are shown in Figure 13. Similar to the observations with MoCo-v2 in Section 4.1, sequential SSL is visibly inferior to joint SSL on streaming data with severe distribution shifts, but sequential SL performs obviously worse than joint SL. In addition, compared with SL models, SSL models show significantly smaller performance gaps between sequential training and joint training.

## B.5   SELF-SUPERVISED TRAINING LOSS OVER TIME

Figure 14 shows the self-supervised training loss at each step in sequential training for various types of streaming data. The loss curves illustrate the average MoCo-v2 training loss for each data chunk, i.e., the contrastive loss. We have two observations. First, given a type of streaming data like the instance incremental sequence, in the sequential training process, the training loss would spike when moving to a new disjoint data chunk. Second, the loss spike value becomes higher with increasing distribution shifts, e.g., training loss increases from instance incremental learning (7.1), random class incremental learning (7.2), to distant class incremental learning (7.4) at the beginning of step 2. The queue size is 65,536 for MoCo-v2, and the training batch size is 256. We record the training loss value averaged across one training batch and plot the loss data point every 250 training iterations, which means that after the first point in the loss curve, we have a filled queue for contrastive learning. Therefore, the loss spike is not because of the training mechanism, such as the queue's refreshing in early training iterations. Instead, it is reasonable to ascribe the loss spike to the distribution shift change of training data, i.e., the new chunk. As for the first chunk, the initial average contrastive

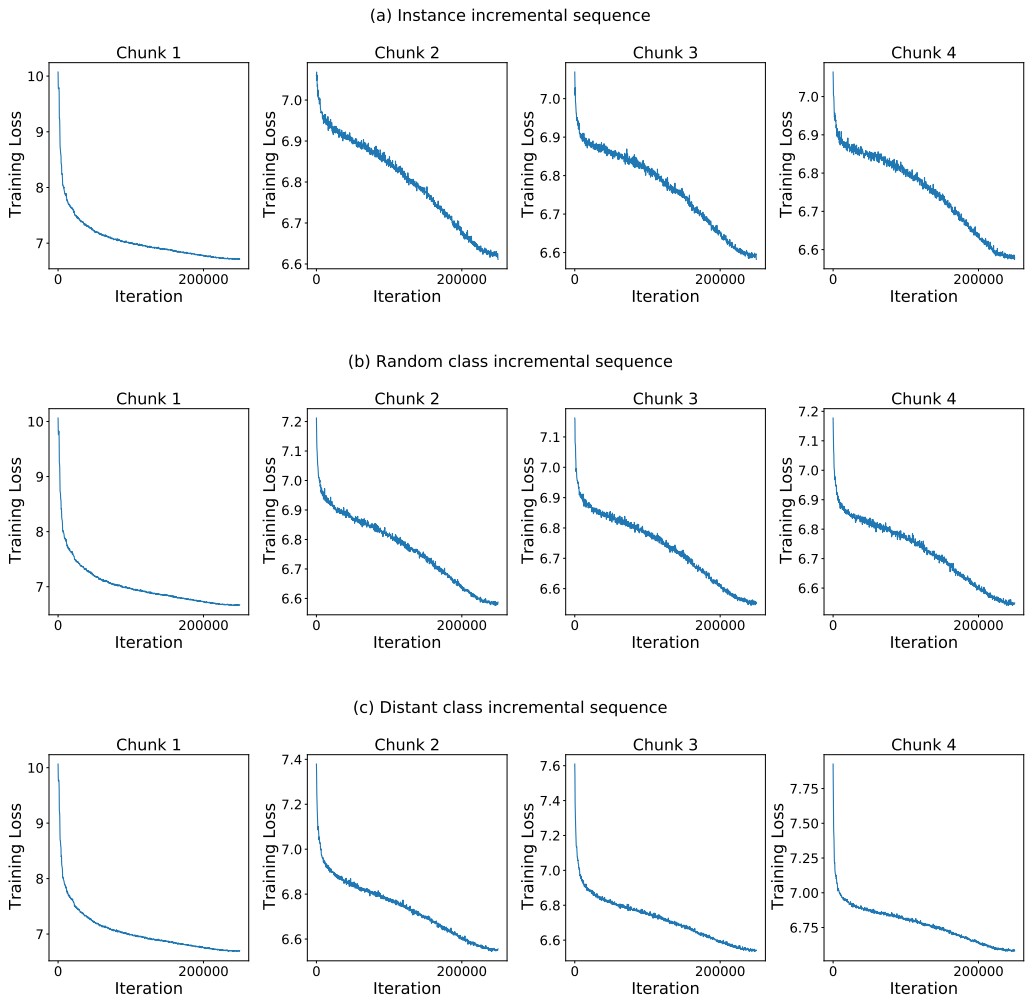

Figure 14: Self-supervised training loss at each step of various types of streaming data.

loss is significant (around 10) because we randomly initialize the representation model. As for subsequent chunks, we inherit the representation model from the previous step. Therefore the initial average contrastive loss is not significant, i.e., around 7.2. On each chunk, we train the model for 200 epochs and finally observe the convergence of contrastive loss. We find that the average contrastive loss would generally converge to the value of around 6.6 for all chunks. We adopt the same training mechanism for each chunk, including the difficulty of the instance discrimination task and hyperparameter settings. We perform contrastive learning until convergence on each chunk. As a result, the final loss value is similar for different chunks across different settings.

## C   MORE EMPIRICAL ANALYSIS

### C.1   UNIFORMITY ANALYSIS OF REPRESENTATIONS.

Uniformity is an important property for good representations (Wang & Isola, 2020). We then compare the uniformity of representations between sequential SL and SSL models. Specifically, we sample images from chunk a and obtain representations $M \in \mathbb{R}^{n \times d}$, where $n$ denotes the number of samples (50,000) and $d$ denotes the dimension of features (2,048). We first use singular values decomposition (SVD) to extract the 2,048 singular values of representations as below:

$$M = U\Sigma V^T, \Sigma \in \mathbb{R}^{n \times d}$$

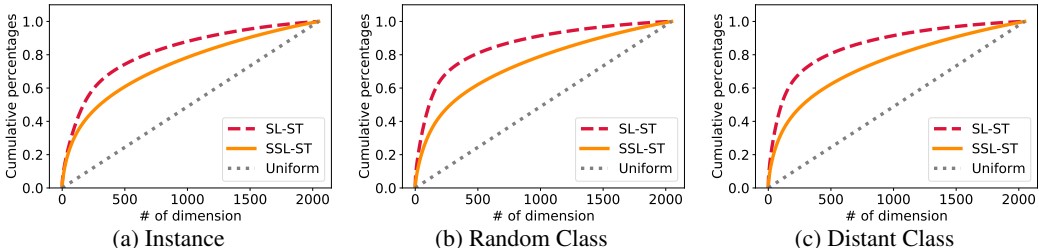

(a) Instance        (b) Random Class        (c) Distant Class

Figure 15: Uniformity of representations. SSL learns more uniform representations than SL.

We first rank the singular values in a descending order and then normalize all singular values to have the sum of 1. We show the cumulative percentages of singular values with increasing dimensions in Figure 15. Our assumption is that representations with more even singular values are more uniform. The corner case is that all singular values are the same, then representations are distributed equally on all dimensions. From results on three types of streaming data, we find that SSL models have more uniform representations than SL models.

## C.2   DETAILS OF BACKWARD TRANSFER AND FORWARD TRANSFER

For accuracy $A^i_{\mathcal{Y}_j}$ on chunk $j$, we first extract the features with the model trained on chunk $i$ for all the examples in the chunk $j$, and then perform KNN classification in the feature space on the chunk $j$. Specifically, we set the number of nearest neighbor k=200 for the KNN classification.

## C.3   CKA SIMILARITY ANALYSIS BETWEEN ST MODELS AND JT MODELS

**CKA similarity.** To further understand the sequential self-supervised pre-training, we then take a closer look at the learned feature representations during the sequential training process. We leverage the linear centered kernel alignment (CKA) (Kornblith et al., 2019a) to measure the similarity of output features between two different representation networks given the same data set as input. If we consider the size of the data set as $n$ and the feature dimension for two networks as $d_1$ and $d_2$, respectively. We use the selected data set to extract features $X \in \mathbb{R}^{n \times d_1}$ from one representation network and features $Y \in \mathbb{R}^{n \times d_2}$ from another representation network. In our experiments, $n$ is 50,000 and both $d_1$ and $d_2$ are 2,048. We first preprocess the two representation matrices by centering the columns. Then the linear CKA similarity between two representations X and Y can be computed as below:

$$CKA(X,Y) = \frac{\|X^T Y\|_F^2}{\|X^T X\|_F^2 \|Y^T Y\|_F^2}.$$

**How are ST models similar to JT models?** We evaluate CKA similarity, for each data chunk, between features from the sequentially trained model and features from the corresponding jointly trained model. The same 50,000 samples are used for CKA features similarity analysis. For example, as shown in Figure 16, at the step of the second data chunk, we compute the CKA similarity value between features of the model jointly trained with the first two data chunks and features from the model sequentially trained after the second data chunk. The corresponding CKA similarity value is 0.4, which indicates for SL, the difference between the ST model and the JT model is very large. In contrast, SSL has a higher similarity of 0.7 between the ST model and the JT model. Particularly, with MAS and data replay, the CKA similarity increases to about 0.9, which means the model trained by sequential SSL extracts nearly the same features as the jointly trained model does. Since the analysis is on the streaming data with severe distribution shifts, this further reinforces our hypothesis that, with the help of suitable continual learning methods, sequential SSL pre-training is promising to replace joint training on streaming data with various distribution shifts.

## C.4   SHARPNESS ANALYSIS

We first provide more details of the sharpness metric in Eq. (2). We compute the sharpness on the respective chunk A for different types of streaming data in ImageNet. Considering that the function $f$ is not differentiable, we sample $\theta'$ from $\mathcal{C}_\epsilon$ and run 50 times to take the minimal accuracy. For

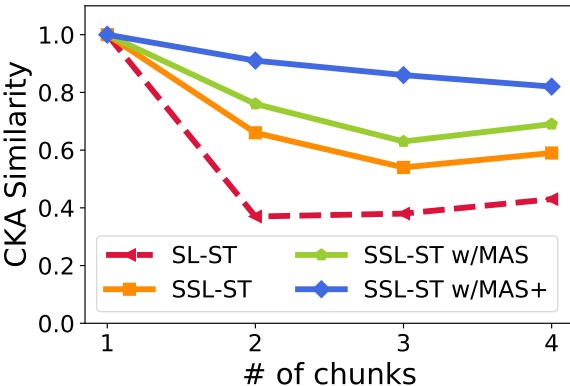

Figure 16: CKA similarity scores between the sequentially trained models and the corresponding jointly trained models at the step of each data chunk on the distant class incremental sequence. Given images in the first data chunk, this figure shows the similarity of features between ST models and the corresponding JT models w.r.t. each data chunk on the distant class incremental sequence. The higher CKA similarity value, the more similar.

computational efficiency, we randomly sample 0.1M data points to perform kNN classification with k=200. We need to clarify that we cannot make comparisons of sharpness values across different sequential settings. Because in different settings, the loss values are calculated on different data samples and the kNN classification tasks are different such as the number of classes to be classified. Therefore, we can only compare the sharpness value between SL and SSL models trained and tested on the same streaming data. Concretely, for the instance incremental sequence, chunk A contains IID samples from ImageNet-1K, classified into 1000 classes. For the random class incremental sequence, chunk A contains 250 random classes from ImageNet-1K, classified into 250 classes. For the distant class incremental sequence, chunk A contains 250 semantically similar classes from ImageNet-1K, classified into 250 classes. We note that in Table 4, the instance incremental sequence has the worst sharpness metric value compared to the other two types of streaming data. This observation is easy to understand. The classification task with 1,000 classes on instance incremental data is more challenging than both classification tasks with only 250 classes on the other two types of streaming data. Therefore, the sharpness value of instance incremental data is the highest. Different from the sharpness experiments, transfer learning experiments evaluate the performance of models on downstream tasks. Therefore, there is no contradiction between the observations from both kinds of experiments.

For flatness visualization, we show the normalized loss along the specified path by performing linearly interpolation between the model after chunk 1 and the model after chunk 2 for different splits, as shown in Figure 17. We can see that compared to SL, loss change is slower for SSL along the linear path, which reflects SSL's superiority in terms of flatness. Specifically, the negative value in the SSL curve means that the performance of SSL continuously improves along the path for instance incremental learning.

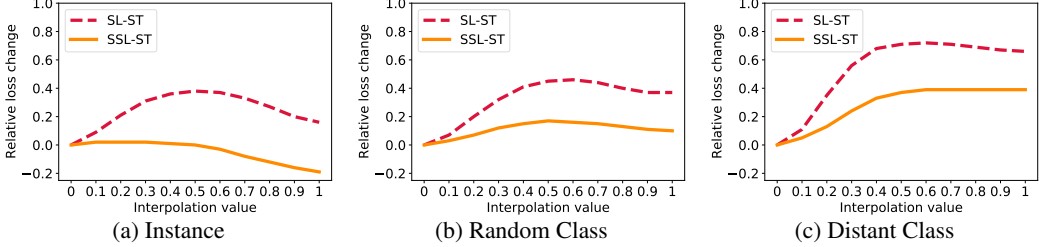

(a) Instance        (b) Random Class        (c) Distant Class

Figure 17: Relation loss change for different interpolations of parameters

