# OpenReview forum: "How Well Does Self-Supervised Pre-Training Perform with Streaming Data?"
_ICLR.cc/2022/Conference — ICLR 2022 Poster_

### Official Review · Reviewer_PFFi · 2021-10-31

**Correctness:** 3
**Technical Novelty And Significance:** 1
**Empirical Novelty And Significance:** 3
**Recommendation:** 6
**Confidence:** 4

**Main Review:**

Strengths
- The paper does a thorough empirical study on different types of streaming data and across a large number of datasets.
- The streaming data setting is important to consider as we train larger pre-trained models and need to update them over time. The new data may or may not have distribution shifts, which this paper shows has an effect on downstream performance vs. re-training the model.
- I am unaware of previous thorough empirical studies on the effectiveness of self-supervised pre-training on streaming data.
- The paper gives interesting observations comparing self-supervised continual learning to supervised continual learning, showing differences under mild distribution shift.
- The results on each individual dataset are plotted clearly. However, the plots do not have error bars.
- The paper resolves some of the issues they see with continual learning using data replay. However, the replay method used seems to make the data size keep increasing - this makes the comparison a bit unfair since the data replay method will get to see more and more data (and the benefit of continual learning diminishes over time). There should be a tradeoff with seeing new and old data.
- The paper also tests another self-supervised method, BYOL.
- The post-hoc analysis of the sequentially trained models shows quite clearly that self-supervised learning suffers less from forgetting for certain types of streaming data (Fig 7).

Weaknesses / Neutral
- It's unclear what changes from a practical point of view here, since the suggestion is to use simple continual learning methods which are known. The main contributions are the observations across different types of streaming data (which we may not have control over?), but these are somewhat alleviated by employing the continual learning methods regardless of the type of streaming data.
- There are other papers out there that seem to consider the streaming / continuing pre-training setting for self-supervised learning (https://openaccess.thecvf.com/content_CVPR_2019/papers/Aljundi_Task-Free_Continual_Learning_CVPR_2019_paper.pdf, https://aclanthology.org/2020.acl-main.740.pdf, https://arxiv.org/abs/2103.12718). Often the story here is that doing more pre-training on a more specialized domain can improve the accuracy on the specialized domain. This is a different point to this paper, which is not looking at the accuracy on specific domains over time (which could also be interesting to keep track of).
- I'm left unsure why there is such a difference between supervised continual learning and self-supervised continual learning. They are definitely measuring different things - with SSL, there is an additional transfer step. Perhaps the finetuning/transfer step takes care of some of the small distribution shift? Some evidence for this: there is a larger gap between streaming and non-streaming when the downstream evaluation is few-shot learning (less finetuning) in Figure 3. Another source of difference could be the loss function - in SSL you need to model all the features whereas in supervised learning, you can focus on features correlated with the label.
- On the setting, there is possible distribution shift across chunks in the streaming data, but it seems that if taken as a whole, the overall distribution is assumed to be stationary in some sense. Perhaps for future work, it may make sense to consider gradual distribution shift over time (of the overall distribution, which changes the relationship between pretraining and downstream).
- In Fig 1, the class semantic trees are not really fully explained. For example, what are the parent nodes? In general the Fig 1 caption could be expanded to more fully specify what the reader should get from the figure. The different types of streaming data are also not explained before they are mentioned in the intro.
- The "instance incremental" setting with IID datapoints, when considered in a huge data setting, is identical to the regular pre-training setup  in some cases - for example, in GPT-3, they didn't even finish one epoch of the data. Thus not having a huge gap isn't too surprising particularly in the large data setting where we make very few passes over the dataset. This paper studies this kind of setting and says that in supervised learning, the gap isn't big (https://arxiv.org/abs/2010.08127). It's interesting that in Figure 11 of your paper, there is a gap between supervised pre-training in a streaming or non-streaming fashion for downstream task performance. This could be an interesting distinction / comparison to make.
- Is there any way to test the claim of "negative transfer" in 4.2? Often, when streaming suffers this "negative transfer", JT does not seem to, even with the same chunks.
- At the bottom of page 3, re-training a model from scratch at every time point is referred to as joint training, but perhaps a more common terminology from online learning is "follow the leader".
- In Figure 7 we only have to show the lower triangular part of the matrix

**Summary Of The Paper:**

The paper considers the setting where data for self supervised pre-training comes in a streaming fashion, where models are incrementally trained on new data. They test 4 different types of streaming data which have different distribution shift properties, on 12 classification datasets and an object detection dataset, using MoCov2 as the self-supervised algorithm. The paper finds that for certain types of streaming data, the downstream task performance of streaming SSL versus "joint" SSL (which is the standard pre-training scheme where all data is available at once) are somewhat similar, while there is a large gap in performance for streaming data types with a large amount of distribution shift. They find that this behavior is unlike the gap between streaming and non-streaming supervised learning, where the gaps are larger in all types of streaming data. Finally, they reduce some of the gaps with data replay and regularization.

**Summary Of The Review:**

The paper provides some interesting observations through pretty thorough experiments on an interesting/important setting for large pre-trained models. They show that continual learning methods can improve the issues they found. There isn't much in the way of explanations of the observations, and the methods aren't novel, but the observations are valuable.

---

> ### Author Response · Authors · 2021-11-23
> **Response to Reviewer PFFi**
>
> Thank you for acknowledging the importance of our proposed method. All your concerns are addressed as follows:
>
> > **Q1**: It's unclear what changes from a practical point of view here since the conclusion is that the gaps between SSL-ST and SSL-JT can be somewhat alleviated by employing continual learning methods regardless of the type of streaming data.
>
> **A1**:  As you have mentioned in Point 6 of “Strength”. There is a tradeoff when applying continual learning methods especially the data replay. Depending on the distribution shift of the practical streaming data, we can adaptively tune the used resources, including time and storage, to conduct the model pre-training. For example, for IID streaming data, the MAS scheme may be enough, while for streaming data with a large distribution shift, data replay may be required.
>
> -------
>
> &nbsp;
>
> > **Q2**: There are other papers out there that seem to consider the streaming [3, 4]/ continuing pre-training setting for self-supervised learning. Often the story here is that doing more pre-training on a more specialized domain can improve the accuracy on the specialized domain. This is a different point to this paper, which is not looking at the accuracy on specific domains over time (which could also be interesting to keep track of).
>
> **A2**: Thanks for pointing out these related works[3, 4]. In this work, we aim to learn generic and transferable downstream-agnostic representation, thus we only use large-scale data to train a representation model in the upstream. Following the settings in popular representation learning works [1,2], all considered downstream tasks or datasets are only for thoroughly evaluating the learned representation.
> Differently, the story in related works like [3, 4] is that after pre-training the model with upstream data, they both keep on pre-training the model with interested downstream data. In this way, they can make the model more adapted to the interested downstream dataset, achieving better downstream transfer. We find this line of works also interesting and meaningful and would like to explore it in the future.
> We have added the discussion with these domain-adaptive pre-training works in related works.
>
>
>
> -------
>
> &nbsp;
>
> > **Q3**: I'm left unsure why there is such a difference between supervised continual learning and self-supervised continual learning. They are definitely measuring different things - with SSL, there is an additional transfer step. Perhaps the finetuning/transfer step takes care of some of the small distribution shift? Some evidence for this: there is a larger gap between streaming and non-streaming when the downstream evaluation is few-shot learning (less finetuning) in Figure 3. Another source of difference could be the loss function - in SSL you need to model all the features whereas in supervised learning, you can focus on features correlated with the label.
>
> **A3**: We have added the discussion of the features learned by SL and SSL models and the discussion on different downstream tasks in Section 4.2. We draw the similar conclusion to yours that SSL models require more fine-tuning work to get better downstream transfer. We also provide more insights and empirical evidence on why SSL forgets less than SL, from the perspective of the sharpness of minima in the loss landscape.
>
> Flat minima in the loss landscape are the minima in which the change of losses is slow in its neighborhood. Note that the models having flat minima in the loss landscape tend to exhibit an impressive generalization ability [5]. When starting with flat minima, we expect that learning new chunks will have a minor effect on the performance of existing chunks, as escaping the wide basin is difficult.
> Therefore, we hypothesize that SSL encourages the model to seek out more flat minima, which increases SSL's resistance to catastrophic forgetting of previous data chunks, further leading to smaller performance gaps between SSL-ST and SSL-JT on downstream tasks.
>
> To verify this hypothesis, we conduct experiments to compare the sharpness of minima between SL and SSL models. Specifically, we measure the sharpness via a standard sharpness metric in [6]. We also conduct model interpolation between sequential models to visualize the linear path in the loss landscape for both SL and SSL models.
> The experimental results in Section 5.3 demonstrate that SSL indeed discovers the more flat minima compared to SL, which verifies our hypothesis and provides an explanation for why SSL suffers less forgetting than SL.
> More details about sharpness analysis can be found in Sec 5.3 and Appendix C.4
> We also add the analysis of the uniformity of the learned representations in Appendix C.1. It shows SSL representations are more uniform.

---

> > ### Author Response · Authors · 2021-11-23
> > **Response to Reviewer PFFi**
> >
> > > **Q4**: On the setting, there is possible distribution shift across chunks in the streaming data, but it seems that if taken as a whole, the overall distribution is assumed to be stationary in some sense. Perhaps for future work, it may make sense to consider gradual distribution shift over time (of the overall distribution, which changes the relationship between pretraining and downstream).
> >
> > **A4**: We believe it is an interesting future direction to explore the behaviors of self-supervised learning on a continual learning setup with gradual distribution shift over time.
> >
> > ------
> >
> > &nbsp;
> >
> >
> > > **Q5**: In Fig 1, the class semantic trees are not really fully explained. For example, what are the parent nodes? In general the Fig 1 caption could be expanded to more fully specify what the reader should get from the figure. The different types of streaming data are also not explained before they are mentioned in the intro.
> >
> > **A5**: We have improved the clarity of Fig 1.  We have added more introduction to the types of streaming data in the revision.
> >
> > ------
> >
> > &nbsp;
> >
> > > **Q6**: The "instance incremental" setting with IID datapoints, when considered in a huge data setting, is identical to the regular pre-training setup in some cases - for example, in GPT-3, they didn't even finish one epoch of the data. Thus not having a huge gap isn't too surprising particularly in the large data setting where we make very few passes over the dataset. This paper studies this kind of setting and says that in supervised learning, the gap isn't big [7]. It's interesting that in Figure 11 of your paper, there is a gap between supervised pre-training in a streaming or non-streaming fashion for downstream task performance. This could be an interesting distinction / comparison to make.
> >
> > **A6**: Firstly, we would like to clarify that our experimental setting is different from the one-pass training setting like GPT-3. In our experiments, we train a model on each data chunk with multiple epochs, i.e., 90 for SL and 200 for SSL.
> > Second, in the paper[7], online learning(streaming learning) can access much more data compared to non-streaming training. In contrast, In our work, sequential learning(streaming learning) is more difficult, because it can only access the current chunk. Besides, joint training(non-streaming learning) allows for access to both previous and current data chunks.
> > As a result, it explains why there is a small gap in our work and why the work [7] believes there is no gap. We appreciate that you find the observation of instance incremental sequence results in Figure 9 interesting. Although the class distribution shift between chunks is negligible, more chunks mean more diverse samples/instances, which is beneficial for good representation learning, we believe it leads to the downstream performance gaps between SL-ST and SL-JT. Furthermore, this nontrivial observation further demonstrates that sequential SSL has less forgetting and better transfer ability than sequential SL, even on the nearly IID streaming data.
> >
> >
> >
> > ---------
> >
> > &nbsp;
> >
> > > **Q7**:  Is there any way to test the claim of "negative transfer" in 4.2? Often, when streaming suffers this "negative transfer", JT does not seem to, even with the same chunks.
> >
> > **A7**:  Based on your suggestions, we provide more empirical evidence to describe and explain the existence of “negative transfer”.
> > We observe severe “negative transfer'' when pre-training models with distant class incremental sequence and evaluating them on Oxford-IIIT Pets. The “negative transfer” exists in two forms (1) ST models may outperform JT models. (2) The model performance may drop with the increase of chunk number for both ST and JT models.
> > To be specific, on the 4-chunk streaming data, at step 2, we find models sequentially trained by the second data chunk show good performance on the testing data of Pets, for both many-shot and few-shot evaluation. In contrast, when the model is sequentially trained by other data chunks, i.e. step 3 and step4, more data for training instead leads to significant performance degradation for sequential training, and visible performance reduction for joint training. We find the reason is that only the second data chunk has lots of similar classes to the Pets dataset.
> >
> > The reason that JT models suffer less “negative transfer” is that JT models can access previous data chunks, except for the current irrelevant chunks. For example, in the sequential learning of D-A-B-C, when training with the B chunk, JT models can have D, A, and B chunks for training. The relevant data in chunk A can significantly mitigate the “negative transfer” brought by data in chunk D and B. However, with only data in chunk B, ST models cannot avoid the severe “negative transfer”.
> > See Section 4.2 and Appendix B.3 for more details of “negative transfer”.

---

> > > ### Author Response · Authors · 2021-11-23
> > > **Response to Reviewer PFFi**
> > >
> > > > **Q8**: At the bottom of page 3, re-training a model from scratch at every time point is referred to as joint training, but perhaps a more common terminology from online learning is "follow the leader
> > >
> > > **A8**: We adopt the commonly used terminology “joint training” from continual learning works [8,9], which allows offline training for incoming data.
> > >
> > > ---------
> > >
> > > &nbsp;
> > >
> > > > **Q9**: Figure 7 we only have to show the lower triangular part of the matrix.
> > >
> > > **A9**: Yes, the CKA similarity matrix is symmetric. We think a complete similarity matrix is easy to catch the required information by directly querying the value located at (row, column).
> > >
> > >
> > > &nbsp;
> > >
> > > [1] A Simple Framework for Contrastive Learning of Visual Representations, ICML 2020
> > >
> > > [2] Momentum Contrast for Unsupervised Visual Representation Learning, CVPR 2020
> > >
> > > [3] Self-Supervised Pretraining Improves Self-Supervised Pretraining, ICCV 2021
> > >
> > > [4] Don’t Stop Pretraining: Adapt Language Models to Domains and Tasks. ACL’20
> > >
> > > [5] On Large-Batch Training for Deep Learning: Generalization Gap and Sharp Minima, ICLR 2016
> > >
> > > [6] Smoothout: Smoothing out sharp minima to improve generalization in deep learning, arxiv preprint, arxiv 2018
> > >
> > > [7] The Deep Bootstrap Framework: Good Online Learners are Good Offline Generalizers ICLR 2021
> > >
> > > [8] Learning a unified classifier incrementally via rebalancing. CVPR 2019
> > >
> > > [9] Uncertainty-guided continual learning with bayesian neural networks. ICLR  2020

---

> > > > ### Comment · Reviewer_PFFi · 2021-11-27
> > > > **Response to authors**
> > > >
> > > > Thanks for the detailed response. In terms of the practical takeaways, does the paper give any prescriptive ways to tune the tradeoff between compute/storage and accuracy if we need data replay? It seems that the data replay in the paper uses more and more data as the number of chunks grows (which seems fine in this paper since there are only 4 chunks), but this would be unsustainable for real streaming data. Perhaps there should be experiments on sequences with much longer than 4 chunks, while using data replay, where it would elicit the need for thinking about how to tune this tradeoff.

---

> > > > > ### Author Response · Authors · 2021-11-27
> > > > > **Response to Reviewer PFFi**
> > > > >
> > > > > Thanks for the prompt response. We are happy to provide further clarifications and discussions on the mentioned data replay strategy.
> > > > >
> > > > > First, we would like to clarify that, as stated in our four takeaways in the introduction, our paper mainly aims to provide a comprehensive study on the sequential self-supervised learning (SSL) with streaming data, including both how and why.
> > > > > As one part of our empirical study, we find that for streaming data with severe distribution shifts, continual learning methods, such as data replay and parameter regularization, can help improve the performance of sequential SSL without visibly reducing the efficiency.
> > > > > We adopt data replay as one possible and easy-to-implement solution to show the benefit of continual learning methods. In addition, we mainly compare the traditional joint pre-training with collective data with our proposed sequential pre-training methods. In sequential training methods with data replay, for simplicity, we add 10\% data of each chunk into the replay buffer [10], which means sequential SSL with data replay is ten times the storage efficiency of joint training methods. We are aware that the size of the replay buffer would increase with more data chunks, but it is still much more storage-saving (tens of times) than the widely used joint pre-training practice.
> > > > >
> > > > > Second, although proposing advanced methods for sequential SSL is not our focus in this paper, we are willing to provide more discussions on the data replay strategy. As for the tradeoff between the accuracy and the compute/storage, access to more previous data contributes to higher accuracy but lower computation (time) efficiency and more storage consumption, as demonstrated in Section 4.5 and Table 1. Especially, keeping all (100\%) previous data in the replay buffer is precisely equivalent to the joint training strategy, which tends to achieve better performance but the worst efficiency. While keeping no previous data (0\%) for data replay is exactly the vanilla sequential training strategy, which achieves the best efficiency but lower performance.
> > > > > We agree with the reviewer that increasing the size of the replay buffer with the growth of data chunks is not sustainable. To make the data replay “sustainable for real streaming data with much longer chunks”, we think setting an upper bound on the size of the replay buffer, such as the number of images, is feasible for realistic scenarios. To be specific, we can determine the upper bound of the replay buffer according to the physical storage limitation.
> > > > > Before the replay buffer approaches the upper bound, the tradeoff is only between the accuracy and training time efficiency, i.e., storing more data (larger ratio) in the replay buffer results in better accuracy but longer training time. When the number of data in the replay buffer exceeds the upper bound, we can discard some old samples to save space for data sampled from new chunks [11].
> > > > > To validate the effectiveness of the bounded memory, we will conduct experiments with 8-chunk ImageNet-21K using a bounded replay buffer.
> > > > > However, because the pre-training experiments with the large-scale dataset are very time-consuming, we can only promise to add them in the final version.
> > > > >
> > > > > To sum up, through the provided simple baselines with data replay or MAS, we hope to call for more attention on proposing more methods to mitigate further the possible downstream performance gaps between sequential SSL and joint SSL with challenging streaming data. If the reviewer has any remaining concerns or questions, please let us know, and we will do our best to address them.
> > > > >
> > > > > [10] Learning a Unified Classifier Incrementally via Rebalancing. CVPR’19
> > > > >
> > > > > [11] iCaRL: Incremental Classifier and Representation Learning. CVPR’17

---

### Official Review · Reviewer_QZfY · 2021-11-02

**Correctness:** 4
**Technical Novelty And Significance:** 2
**Empirical Novelty And Significance:** 3
**Recommendation:** 5
**Confidence:** 4

**Main Review:**

Strength
1.	This paper firstly investigates the sequential training situation for self-supervised learning, which is an important problem for the widespread use of self-supervised learning in real world application. They also propose some scenarios for training model with streaming data.
2.	The experimental results show that the problem of catastrophic forgetting due to training with streaming data is less problematic for self-supervised pretraining compared to the supervised pretraining, for all scenarios.
3.	They show that some existing continual learning techniques are effective for self-supervised sequential pretraining.

Weakness
1.	The novelty and technical contribution are limited. Although this paper proposes some scenarios for sequential training with streaming data and shows that the self-supervised learning methods work better than supervised learning baselines under these scenarios, it is hard to find further contribution except this empirical discovery
2.	It seems that the analysis results in Section 5 are not very helpful for understanding. For example, in subsection 5.1, there is no explanation what BWT and FWT mean and how this is related to the catastrophic forgetting issue. Also, it seems that the result of BWT, FWT, CKA and feature reconstructions are the natural result of less catastrophic forgetting, and provide meaningful insights to make a further improvement on sequential self-supervised pretraining.


**Summary Of The Paper:**

This paper explores a more realistic situation for self-supervised representation learning of training with streaming data. In this paper, four different situations for training with streaming data are investigated. The experimental results show that the self-supervised pretraining less suffers from catastrophic forgetting compared to the supervised pretraining, when the model is trained sequentially using streaming data. The experimental results also show that existing continual learning techniques are also effective for self-supervised learning with streaming data.

**Summary Of The Review:**

This paper empirically studied on sequential self-supervised learning problem, which is never done before, and show that self-supervised learning is more robust to catastrophic forgetting problem compared to supervised learning. Although exploring the problem of sequential self-supervised learning is meaningful, it is hard to find insights beyond empirical results to make improvement on the top of the discovery of this work in this paper. As a result, I prone to reject this paper. However, I am open to any objection and further discussion, if the authors make their point and insight of this paper clearer.

---

> ### Author Response · Authors · 2021-11-23
> **Response to Reviewer QZfY**
>
> Thanks for your valuable suggestions. We address all your concerns as follows
>
> > **Q1**: The novelty and technical contribution are limited. Although this paper proposes some scenarios for sequential training with streaming data and shows that the self-supervised learning methods work better than supervised learning baselines under these scenarios, it is hard to find further contribution except this empirical discovery
>
> **A1**: The empirical discovery from extensive experiments is our first contribution and we have clarified our contributions in the revision paper, as stated in the last paragraph of Introduction. We argue that our contributions are far beyond the empirical discovery.  Specifically, we have four main contributions or takeaways:
>
>
> (1). We find the following observations based on extensive pre-training and downstream experiments. Sequential SSL models exhibit the on-par transfer learning performance as joint SSL models on streaming data with negligible or mild distribution shift. As for streaming data with severe distribution shifts or longer sequences, i.e., the distant class incremental sequence, evident performance gaps exist between sequential SSL and joint SSL models. Such performance gaps, however, can be mitigated effectively and efficiently with unsupervised parameter regularization and simple data replay.
>
> (2). Based on the above finding, we conclude that the standard joint training paradigm may be unnecessary for SSL pre-training. Instead, sequential SSL is performance-competitive but more time-efficient and storage-saving and is well worth considering as the practical practice for self-supervised pre-training with streaming data.
>
> (3). Compared with supervised learning (SL) models, SSL models consistently show smaller performance gaps between ST and JT. Inspired by such observations, we conduct comprehensive analysis on the knowledge forgetting behavior,  including the KNN classification accuracy-based backward/forward transfer, CKA similarity-based representations similarity, and image reconstructions from features. The comprehensive investigation of learned representations demonstrates that sequential SSL models are less prone to catastrophic forgetting than SL models.
>
> (4). Through the empirical analysis on the sharpness of minima in the loss landscape, we find that SSL models have wider minima than SL models, which we hypothesize is the reason for less forgetting of SSL models.
>
> In addition, we would like to further clarify the significance of our paper.
>
> First, our work would benefit the self-supervised learning community. Almost all previous works focus on how to improve the SSL models pre-trained on collective large-scale datasets, such as ImageNet, YFCC100M, and Instagram-1B. Differently, we first take a step to apply SSL pre-training under realistic data collection scenarios where unlabeled data is streaming with various distribution shifts. We also provide the first comprehensive empirical study on this realistic and promising pre-training setting, supported by a large number of experiments. We are glad that the reviewer also appreciates this in the first point of “Strengths.”
> Besides, as acknowledged by Reviewer 4UNG in the first point of “Strengths”, our work “provides a comprehensive benchmark for the community to explore the setting of sequential self-supervised learning in further detail.”
>
> Second, our work would also benefit the continual learning community. Our comprehensive analysis on SSL and SL representations uncover that SSL suffers less catastrophic forgetting than SL. In addition, SSL has both better negative and forward transfer than SL. Moreover, we provide a justified hypothesis to explain the different forgetting behavior between SL and SSL. Current continual learning works are struggling with catastrophic forgetting in supervised learning tasks. Therefore, we believe our comprehensive analysis on forgetting is especially beneficial to the continual learning community.

---

> > ### Author Response · Authors · 2021-11-23
> > **Response to Reviewer QZfY**
> >
> > > **Q2**: It seems that the analysis results in Section 5 are not very helpful for understanding. For example, in subsection 5.1, there is no explanation what BWT and FWT mean and how this is related to the catastrophic forgetting issue.
> >
> > **A2**: Thanks for your valuable feedback on BWT and FWT. We respectfully disagree with the opinion that “the analysis results in Section 5 are not very helpful for understanding.”
> >
> > We would like to argue that the analysis in Section 5 is “particularly insightful”(Reviewer 4UNG), especially insights on forgetting such as the relationship between forgetting and distribution shift, example forgetting, SSL suffers less forgetting, etc.
> > Without Section 5, what we know about SSL and SL is that compared with supervised learning (SL) models, SSL models consistently show smaller performance gaps on the downstream tasks between ST and JT. We cannot probe such differences between SL and SSL to their different forgetting behavior. As stated in “summary” in Section 4, “The above difference between SL and SSL models motivates us to further investigate the forgetting property in Section 5.” Therefore, we argue that our comprehensive analysis on the forgetting behavior as well as the added hypothesis are especially helpful for understanding sequential SSL.
> >
> > As for backward transfer (BWT) and forward transfer (FWT), we add more explanations in Section 5.1. BWT and FWT are the standard evaluation behavior in supervised continual learning [1] used to assess the knowledge transfer in sequential learning.  Backward transfer refers to the improvement of performance on previously learned chunks when learning new chunks, where large negative transfer is also known as catastrophic forgetting.
> > Forward transfer measures the improvement in performance on the novel chunk with the accumulation of knowledge from previous chunks.
> > Through the extensive analysis on BWT and FWT, we can obtain the following observations about forgetting:
> >
> > (1) Learning method: SSL itself is less prone to catastrophic forgetting than SL, especially that SSL achieves positive backward transfer on instance incremental sequence. It illustrates that SSL is more suitable for streaming data;
> >
> > (2) Types of streaming data: The model suffers progressively severe forgetting when the distribution shift increases for both SSL and SL cases.
> >
> > (3) Example forgetting: It is observed that forgetting is less severe in top-5 classification than top-1 classification, which indicates that the knowledge is not fully forgotten.

---

> > > ### Author Response · Authors · 2021-11-23
> > > **Response to Reviewer QZfY**
> > >
> > > > **Q3**: Also, it seems that the result of BWT, FWT, CKA and feature reconstructions are the natural result of less catastrophic forgetting, and provide meaningful insights to make a further improvement on sequential self-supervised pertaining.
> > >
> > > **A3**: Thanks for your feedback. We would like to respectfully disagree with the opinion that “the result of BWT, FWT, CKA and feature reconstructions are the natural result of less catastrophic forgetting”. Because, before Section 5, we can only draw the conclusion that the observation that SSL shows smaller performance gaps on the downstream tasks or datasets between ST and JT models, compared with SL. However, the forgetting analysis in Section 5 is completely conducted on the upstream data, i.e., using pre-training data chunks.
> > > To our best knowledge, there is no demonstrated relationship between the downstream performance and the upstream performance. In other words, a smaller performance gap on the downstream tasks between ST and JT models does not imply less forgetting with pre-training data. Therefore, we cannot conclude the forgetting behavior of both SL and SSL models, depending on experiments before Section 5.
> > >
> > > In Section 5, we analyze the knowledge forgetting of previous tasks to give more insights from two perspectives. Following the work in continual learning [1], we evaluate the transfer ability of both SL and SSL representations via the standard backward and forward transfer analysis in continual learning . Following another empirical study paper in continual learning [2], we conduct CKA analysis among sequentially trained models, we directly analyze the forgetting of learned representations in the process of sequential learning. Following [3], we visualize the images reconstruction by feature inversion to directly analyze the information included in the learned representations. Last but not least, we provide our empirically justified hypothesis for why SSL models forget less than SL models.
> > >
> > > Note FWT has nothing to do with the forgetting of previous chunks, but it measures the improvement in performance on the novel chunk with the accumulation of knowledge from previous chunks.Therefore, insights with FWT are not natural results of catastrophic forgetting. Through experiments of images reconstruction by features inversion, we have similar observations that SSL representations can capture more detailed information about the rwa image than SL representations, which is consistent with [3]. This insight is also not natural.
> > >
> > > Therefore, we would like to clarify that our analyses, including BWT & FWT, CKA, and feature reconstructions, are non-trivial, meaningful, and insightful.
> > >
> > > [1] Gradient episodic memory for continual learning, NeurIPS 2017
> > >
> > > [2] Anatomy of Catastrophic Forgetting: Hidden Representations and Task Semantics, ICLR 2021
> > >
> > > [3] What makes instance discrimination good for transfer learning? ICLR 2021

---

### Official Review · Reviewer_RJbi · 2021-11-02

**Correctness:** 4
**Technical Novelty And Significance:** 4
**Empirical Novelty And Significance:** 4
**Recommendation:** 8
**Confidence:** 4

**Main Review:**

Pros:
    1. The submission is well grounded. The task studied in the manuscript is highly practical. the motivation of conducting such an analysis is strong, since the incremental learning setup is indeed practical in many real-world scenarios.
    2. The amount of experiments is enormous and sufficient to draw the conclusion.
    3. The experimental designs, especially the four categories of incremental learning setups, are interesting and to the point.
    4. The manuscript is well organized and well written, especially the abstract and introduction that gives a good tutorial.

I have some concerns and comments as follows:
    1. Please elaborate on the rationale behind the design of the four scenarios, as early as in the introduction. In other words, why these four categories but nothing in-between or beyond?
    2. It would be good if the authors can provide more insights and potential applications apart from revealing the experimental observations. In other words, what message of the experimental results convey?
    3. Intuitively, why do self-supervised models forget less than supervised models?
    4. Given the performances of the downstream tasks, any insights one can have? Like the affinities across tasks?


**Summary Of The Paper:**

The authors of this submission worked on an interesting and quite practical task, i.e., to study the self-supervised behavior under the continual learning setup. To this end, the authors studied four types of incremental-learning settings, i.e., Instance incremental sequence, Random class incremental sequence, Distant class incremental sequence, Domain incremental sequence. The four types are designed to reflect different incremental-learning scenarios with different semantics.The authors have also conducted extensive empirical validations, including pre-training >500 models on four categories of pre-training streaming data from ImageNet and DomainNet, and evaluating them on three types of downstream tasks and 12 different downstream datasets.

**Summary Of The Review:**

Th authors focus on an interesting and practical task. Prior works have largely overlooked this task and the proposed work is the first to explore along this line. The motivation of conducting such an analysis is strong, since the incremental learning setup is indeed practical in many real-world scenarios. Extensive experiments are also conducted to support the proposed method, which are very solid.

---

> ### Author Response · Authors · 2021-11-23
> **Response to Reviewer RJbi**
>
> Thank you for acknowledging the significance of our work. Based on your comments, we would like to further address your concerns as follows:
>
> > **Q1**: Please elaborate on the rationale behind the design of the four scenarios, as early as in the introduction. In other words, why these four categories but nothing in-between or beyond?
>
> **A1**: We have added more introduction to the rationale behind the proposed streaming data in the introduction. We would like to clarify that we have two main considerations behind the proposed four types of streaming data.
>
> The main consideration is to mimic common practical data collection scenarios where streaming data may include different levels of data distribution shift. To this end, we proposed four types of streaming data targeting for four degrees of commonly-seen distribution shift in realistic scenarios.
>
> The second consideration is for the ease of a comprehensive empirical study on pre-training with streaming data. For the first time to provide such a systematic study, we believe using streaming data with various degrees of distribution shifts is beneficial for the community to have a comprehensive understanding of the sequential pre-training problem. Streaming data with other complex distribution shifts may be feasible, e.g., mixing different distribution shifts within one streaming data sequence may be more realistic. We call for more attention on pre-training with streaming data and further extending the types of considered data sequence wherein. But for our work, a study on streaming data containing mixed distribution shifts is not easy to probe into the properties of the studied problem. Therefore, we adopt the design of the four scenarios in our paper.
>
> --------
>
> &nbsp;
>
> > **Q2**: It would be good if the authors can provide more insights and potential applications apart from revealing the experimental observations. In other words, what message of the experimental results convey?
>
> **A2**: We provide more insights behind the experimental observations. Especially, as would be mentioned in A3, we have added a justified hypothesis with consistent experimental results to explain why SSLmodels  forget less than SL models.
> Through the first comprehensive empirical study on pre-training with streaming data, we aim to call for more attention for the much more efficient sequential SSL-based pre-training. Specifically, we have four main takeaways to convey based on our experiments:
>
> (1). Sequential SSL models exhibit the on-par transfer learning performance as joint SSL models on streaming data with negligible or mild distribution shift. As for streaming data with severe distribution shifts or longer sequences, i.e., the distant class incremental sequence, evident performance gaps exist between sequential SSL and joint SSL models. Such performance gaps, however, can be mitigated effectively and efficiently with unsupervised parameter regularization and simple data replay.
>
> (2). Based on the above finding, we conclude that the standard joint training paradigm may be unnecessary for SSL pre-training. Instead, sequential SSL is performance-competitive but more time-efficient and storage-saving and is well worth considering as the practical practice for self-supervised pre-training with streaming data.
>
> (3). Compared with supervised learning (SL) models, SSL models consistently show smaller performance gaps between ST and JT. Inspired by such observations, we conduct comprehensive analysis on the knowledge forgetting behavior,  including the KNN classification accuracy-based backward/forward transfer, CKA similarity-based representations similarity, and image reconstructions from features. The comprehensive investigation of learned representations demonstrates that sequential SSL models are less prone to catastrophic forgetting than SL models.
>
> (4). Through the empirical analysis on the sharpness of minima in the loss landscape, we find that SSL models have wider minima than SL models, which we hypothesize is the reason for less forgetting of SSL models.
>
> As for potential applications, we believe that our work would provide a more efficient yet effective way of pre-training with daily-generated large-scale unlabeled data in practical industrial applications. Also, the good knowledge forgetting behavior of SSL models and the comprehensive analysis of sharpness of minima in the loss landscape would benefit interesting future works in continual learning.

---

> > ### Author Response · Authors · 2021-11-23
> > **Response to Reviewer RJbi**
> >
> > > **Q3**: Intuitively, why do self-supervised models forget less than supervised models?
> >
> > **A3**: We provide more insights and empirical evidence on why SSLmodels  forget less than SL models, from the perspective of the sharpness of minima in the loss landscape.
> >
> > Flat minima in the loss landscape are the minima in which the change of losses is slow in its neighborhood. Note that the models having flat minima in the loss landscape tend to exhibit an impressive generalization ability [1]. When starting with flat minima, we expect that learning new chunks will have a minor effect on the performance of existing chunks, as escaping the wide basin is difficult.
> > Therefore, we hypothesize that SSL encourages the model to seek out more flat minima, which increases SSL's resistance to catastrophic forgetting of previous data chunks, further leading to smaller performance gaps between SSL-ST and SSL-JT on downstream tasks.
> >
> > To verify this hypothesis, we conduct experiments to compare the sharpness of minima between SL and SSL models. Specifically, we measure the sharpness via a standard sharpness metric in [2]. We also conduct model interpolation between sequential models to visualize the linear path in the loss landscape for both SL and SSL models.
> > The experimental results in Section 5.3 demonstrate that SSL indeed discovers the more flat minima compared to SL, which verifies our hypothesis and provides an explanation for why SSL suffers less forgetting than SL.
> > More details about sharpness analysis can be found in Sec 5.3 and Appendix C.4
> > -------
> >
> > &nbsp;
> >
> > > **Q4**: Given the performances of the downstream tasks, any insights one can have? Like the affinities across tasks?
> >
> > **A4**: As for insights from the performance of different downstream tasks. We mainly have two interesting observations and provide the corresponding explanations to them in Section 4.2.
> >
> > As for different downstream tasks, we observe SSL models perform competitively to SL-JT models under many-shot evaluation while SSL models significantly underform SL-JT models under few-shot evaluation. The underlying insight is that supervised features are correlated with labels and more discriminative, thus easy to directly transfer to downstream datasets similar to upstream pre-training data (DomainNet or ImageNet). For example, SL models dominate most few-shot object or scene classification tasks but fail on DTD dataset, a texture classification dataset which shares no common classes with ImageNet or DomainNet. In contrast, self-supervised features are more generalized and comprehensive, thus requiring more fine-tuning for desirable downstream transfer.
> >
> > As for evaluation with different downstream datasets, we find the sequentially trained model may suffer “negative transfer” on some specific downstream datasets. A concrete example is given in Appendix B.3. The insight behind pre-training with more data chunks does not necessarily benefit a specific downstream dataset if the added training data are irrelevant to the downstream dataset.
> >
> > It is a great suggestion on the analysis of the affinities across tasks to further understand the downstream tasks. While in this work, we aim to provide a comprehensive study on pre-training with upstream data. Therefore our explorations or ablations mainly focus on model pre-training. As for the downstream tasks, we only use them to evaluate the pre-trained models, following popular pre-training works [3, 4]. We thus leave the study of downstream tasks for future work.
> >
> > [1] On Large-Batch Training for Deep Learning: Generalization Gap and Sharp Minima, ICLR 2016
> >
> > [2] Smoothout: Smoothing out sharp minima to improve generalization in deep learning, arxiv preprint, arxiv 2018
> >
> > [3] A Simple Framework for Contrastive Learning of Visual Representations, ICML 2020
> >
> > [4] Momentum Contrast for Unsupervised Visual Representation Learning, CVPR 2020

---

### Official Review · Reviewer_4UNG · 2021-11-05

**Correctness:** 3
**Technical Novelty And Significance:** 2
**Empirical Novelty And Significance:** 3
**Recommendation:** 6
**Confidence:** 4

**Main Review:**

Strengths

1. The paper provides an extensive framework for assessing the downstream task performance of sequential self-supervised learning through the four different settings of increasing severity of data distribution drift. I believe this framework unifies the different settings considered by previous works in this domain and provides a comprehensive benchmark for the community to explore the setting of sequential self-supervised learning in further detail.

2. The paper considers downstream performance on several standard benchmark datasets that contributes to the generalization of the insights and analysis to other vision datasets. There is also analysis on both many-shot and few-shot classification tasks which is essential to cover the diverse range of downstream tasks found in practice.

3. Understanding the performance gap between SSL-ST and SSL-JT at different degree of distribution drift allows to identify what level of distribution drift hurts the performance. I find this analysis particularly insightful. Moreover, the paper also experiments with some continual learning approaches to try mitigating this performance reduction. This information can be useful to understand in what type of sequential self-supervised training scenario, there may be a need to experiment with continual learning approaches.

4. The analysis of feature similarity helps to understand that in self-supervised learning, sequential setting behaves quite similar to joint setting in terms of learned feature representations.

5. The additional analysis on space-time efficiency and backward/forward transfer analysis of performing sequential self-supervised learning is also quite extensive and useful to understand the benefits of the approach.

Weakness
1. It seems that the paper overall has little to no insight of its own to offer on why the sequential self-supervised learning is much closer in performance to joint self-supervised learning. In some places (Sec 4.2 and Sec 6), the paper does provide some explanation to this behavior but that is (1) conjectured in that there are no experiments conducted or theoretically justified if that is indeed the rationale and (2) cited from other papers. The paper does provide empirical evidence on how well SSL-ST is doing compared to SSL-JT but that seems more of an outcome of how the downstream tasks are performing in the two settings and not exactly shedding light on what is making SSL-ST do so well compared to SSL-JT while SL-ST is much worse than SL-JT. I would suggest the authors to add more empirical evidence about why SSL-ST is not so much worse than SSL-JT across the different sequential settings considered. Further, there can be more empirical evidence added to the claims that there is ‘negative transfer’ happening (Sec 4.2) or the features learned by uniformly distributed over feature space (Sec 7) and this is indeed a potential reason for the trends observed.

2. Continual learning settings are very sensitive to the empirical setup. Changes such as the number of classes per chunk and the size of the replay buffer can make a drastic change in the performance. I am not very clear why the sequential setting has just 4 chunks. I would suggest the authors to experiment on much longer sequences such as 10 or 20 chunks. Understanding the impact of having longer sequences is crucial to assess the full extent of forgetting and negative impact of distribution drift. Moreover, tasks like self-supervised learning are only practical when leveraging extremely large amounts of data as shown with Instagram 1B (Mahajan et al, 2018). Such works are already instance incremental and at large scales, having more than 4 chunks will be highly likely.

3. The experiment details lack some important details to validate the soundness of the setup in which experiments were performed. For eg, when using MoCo-v2, what is the size of the queue used for training? Is the queue refreshed before training on a new chunk? Or is it simply preserved and enqueued/dequeued as usual? In the case of the latter, my concern will be that the setup is not really working on disjoint chunks and the observations may not be valid. Further, if possible, I would request the authors if they could provide the training loss curves for the different settings across the chunks. I would expect the training loss to spike whenever the training proceeds to a new disjoint chunk and the spikes should be higher for higher severity of data distribution drift. If this is not the case, there is a likelihood of a leakage between chunks which affects the sanity of the sequential setup.

4. I am curious why there is a drop in SSL-ST wrt SSL-JT in figure 11 for instance incremental sequence setting. In Appendix B (Pretraining), it says that ‘we consider one random sequence as the data are randomly divided’. Does this mean that the way SSL-ST is implemented in an instance incremental setting, it is identical to SSL-JT? Please correct me if I am wrong. But if this is correct, shouldn’t the two curves have no performance difference?

**Summary Of The Paper:**

This paper studies how models perform on downstream image classification tasks when they are trained via self-supervision using streaming data. Streaming data refers to a sequential training setting  (ST) where the entire training data is not available for training at the same time but can be only used in disjoint chunks. When training on one chunk of data, the previous chunks are unavailable for training during this sequential training. When the entire training data is available for training at the same time, it is called joint training (JT). The key insight of the work is that in the presence of little to moderate drift in the data distribution across chunks, the downstream task performance of self-supervised training in sequential setting (SSL-ST) is comparable to that of self-supervised training in joint setting (SSL-JT). While in the presence of severe drift in data distribution across chunks, SSL-ST performs worse than SSL-JT but through the application of continual learning approaches like data replay and MAS (Aljundi et al, 2018), this performance gap can be reduced to a large extent. Moreover, when training is done in a supervised manner (SL), sequential setting (SL-ST) performs much worse than joint setting (SL-JT)

The work experiments on four kinds of sequential settings with varying degrees of data distribution drift across consecutive chunks - (1) instance incremental sequence, where each chunk of unlabeled data is derived from the same set of classes having i.i.d. samples in each chunk, (2) random class incremental sequence, where data in each chunk belongs to a different set of randomly chosen classes, (3) distant class incremental sequence, where each chunk belongs to a different set of classes such that there is a larger semantic gap between the set of classes and (4) domain incremental sequence, where the chunks belong to different domains thereby exhibiting severe distribution shift. For the first three settings, Imagenet-1k is used which is split into 4 chunks as per the setting. For the fourth setting, DomainNet is used. MoCo-v2 is primarily used in the empirical analysis along with some experiments on BYOL are also conducted to show that the trends are observed in other SSL approaches as well.

The work also conducts an analysis on the amount of forgetting using the backward and forward transfer analysis which shows that compared to supervised setting where there is significant amount of catastrophic forgetting, self-supervised sequential setting has much lower forgetting. Further analysis via Centered Kernel Alignment (CKA) suggests that compared to supervised setting, SSL-ST has a higher feature similarity between two sequential models. Even the feature similarity of SSL-ST with SSL-JT is higher than the corresponding similarity of SL-ST with SL-JT.

**Summary Of The Review:**

I would like to recommend the paper for ‘marginally above the acceptance threshold’ rating. The authors have addressed some of my concerns. Some of my concerns still exist such as how concretely the new analysis using sharpness of loss minima justifies SSL-ST to be compared to SSL-JT. I also have some concerns on the soundness of the loss curve and whether the rise in the loss is due to distribution drift or just recalibration of the SSL queue. Although there is certainly areas to improve, the paper does provide an interesting perspective to compare SSL and SL and also support it with different kinds of analysis that led me to revise the rating.

---

> ### Author Response · Authors · 2021-11-23
> **Response to Reviewer 4UNG**
>
> Thank you for the valuable suggestions. We carefully address your concerns as follows:
> > **Q1** : It seems that the paper overall has little to no insight of its own to offer on why the sequential self-supervised learning is much closer in performance to joint self-supervised learning. The paper does provide empirical evidence on how well SSL-ST is doing compared to SSL-JT but that seems more of an outcome of how the downstream tasks are performing in the two settings and not exactly shedding light on what is making SSL-ST do so well compared to SSL-JT while SL-ST is much worse than SL-JT. I would suggest the authors to add more empirical evidence about why SSL-ST is not so much worse than SSL-JT across the different sequential settings considered.
>
> **A1**: We provide more insights and empirical evidence on why SSL forgets less than SL in revision, from the perspective of the sharpness of minima in the loss landscape.
>
> Flat minima in the loss landscape are the minima in which the change of losses is slow in its neighborhood. Note that the models having flat minima in the loss landscape tend to exhibit an impressive generalization ability [1]. When starting with flat minima, we expect that learning new chunks will have a minor effect on the performance of existing chunks, as escaping the wide basin is difficult.
> Therefore, we hypothesize that SSL encourages the model to seek out more flat minima, which increases SSL's resistance to catastrophic forgetting of previous data chunks, further leading to smaller performance gaps between SSL-ST and SSL-JT on downstream tasks.
>
> To verify this hypothesis, we conduct experiments to compare the sharpness of minima between SL and SSL models. Specifically, we measure the sharpness via a standard sharpness metric in [2]. We also conduct model interpolation between sequential models to visualize the linear path in the loss landscape for both SL and SSL models.
> The experimental results in Section 5.3 demonstrate that SSL indeed discovers the more flat minima compared to SL, which verifies our hypothesis and provides an explanation for why SSL suffers less forgetting than SL.
> More details about sharpness analysis can be found in Sec 5.3 and Appendix C.4
> -------------------------
> &nbsp;
>
> > **Q2** : In some places (Sec 4.2 and Sec 6), the paper does provide some explanation to this behavior but that is (1) conjectured in that there are no experiments conducted or theoretically justified if that is indeed the rationale and (2) cited from other papers. Further, there can be more empirical evidence added to the claims that there is ‘negative transfer’ happening (Sec 4.2) or the features learned by uniformly distributed over feature space (Sec 7) and this is indeed a potential reason for the trends observed.
>
> **A2**: Thanks for your very insightful feedback. As mentioned in A1, we have added a justified hypothesis with consistent experimental results to explain the behavior.
> In addition, based on your suggestions, we provide more empirical evidence to describe and explain the existence of “negative transfer” and to compare the uniformity of feature space between supervised models and self-supervised models.
>
> We observe severe “negative transfer'' when pre-training models with distant class incremental sequence and evaluating them on Oxford-IIIT Pets. The “negative transfer” exists in two forms (1) ST models may outperform JT models. (2) The model performance may drop with the increase of chunk number for both ST and JT models.
> To be specific, on the 4-chunk streaming data, at step 2, we find models sequentially trained by the second data chunk show good performance on the testing data of Pets, for both many-shot and few-shot evaluation. In contrast, when the model is sequentially trained by other data chunks, i.e. step 3 and step4, more data for training instead leads to significant performance degradation for sequential training, and visible performance reduction for joint training. We find the reason is that only the second data chunk has lots of similar classes to the Pets dataset.
> See Section 4.2 and Appendix B.3 for more details of “negative transfer”.
>
> We also show the uniformity of representations from both SL and SSL models. Specifically, we simply compare the distribution of the singular values between SL and SSL representations. We observe that SSL representations have more uniform singular values than SL representations. We note [3] has both theoretically and empirically shown that contrastive loss has a uniformity regularization on the feature space, which is consistent with our empirical analysis. See Appendix C.1 for more details about the uniformity.

---

> > ### Author Response · Authors · 2021-11-23
> > **Response to Reviewer 4UNG**
> >
> > > **Q3**: Continual learning settings are very sensitive to the empirical setup. Changes such as the number of classes per chunk and the size of the replay buffer can make a drastic change in the performance. I am not very clear why the sequential setting has just 4 chunks. I would suggest the authors to experiment on much longer sequences such as 10 or 20 chunks. Understanding the impact of having longer sequences is crucial to assess the full extent of forgetting and negative impact of distribution drift. Moreover, tasks like self-supervised learning are only practical when leveraging extremely large amounts of data as shown with Instagram 1B (Mahajan et al, 2018). Such works are already instance incremental and at large scales, having more than 4 chunks will be highly likely.
> >
> > **A3**: Thank you for the valuable suggestion on experiments with longer sequences. We would like to clarify that to ensure each data chunk has enough unlabeled data for effective SSL representation learning, we set the chunk length as 4 rather than a larger number for three types of ImageNet-based streaming data. As for the DomainNet-based streaming data, we directly take 5 domains as 5 separate data chunks.
> >
> > We agree that experiments on much longer sequences are required and thanks for the suggestion of using Instagram 1B. Since our main experiments are on ImageNet-1k, here we use another large-scale image dataset, i.e. ImageNet-21k, to conduct SSL-ST & SSL-JT with an instance incremental sequence with 8 chunks. To obtain the 8-chunk IID data sequence, except the 1,000 classes in ImageNet-1k, we sample another 1,000 classes of images from ImageNet-21k, mix them with ImageNet-1k data, and then randomly divide all the 2.4 M data into 8 chunks. Then, we adopt the same training of SSL-ST or SSL-JT for this longer data sequence.
> >
> > Limited by computing, we only obtain results of SSL and will add results of SL on this longer sequence in the future. Results show that the longer sequence does increase the performance gaps between ST models and JT models. But we find the gaps are not significant and believe continual learning methods discussed in the main text can effectively mitigate such gaps. See Section 4.1 and Appendix B.2 for more details of the realistic longer sequence.
> >
> > --------------------
> >
> > &nbsp;
> >
> > > **Q4**: The experiment details lack some important details to validate the soundness of the setup in which experiments were performed. For eg, when using MoCo-v2, what is the size of the queue used for training? Is the queue refreshed before training on a new chunk? Or is it simply preserved and enqueued/dequeued as usual? In the case of the latter, my concern will be that the setup is not really working on disjoint chunks and the observations may not be valid. Further, if possible, I would request the authors if they could provide the training loss curves for the different settings across the chunks. I would expect the training loss to spike whenever the training proceeds to a new disjoint chunk and the spikes should be higher for higher severity of data distribution drift. If this is not the case, there is a likelihood of a leakage between chunks which affects the sanity of the sequential setup.
> >
> > **A4**:  Thanks for pointing out the missing important details on experiments. We make clarifications point by point. As for the queue in MoCo-v2 training, we adopt the same as the original MoCo-v2, i.e. the queue size is 65,536. For data chunks fewer than 65,536 like DomainNet chunks, we store all samples of the current chunk for contrastive learning. In sequential training, the queue is refreshed before training on a new chunk. We never preserve previous data in the queue,  to ensure the model training is sequentially conducted on disjoint chunks. For the possibly used data replay strategy, another special replay buffer is used to preserve a small fraction of data from each previous chunk, like 10% in our experiments. We have added these details in Appendix A.2.
> >
> > As for the training loss, we have added training loss curves of SSL-ST for three ImageNet-based streaming data in Appendix B.5. We have observations absolutely consistent with your expectations, i.e., the training loss would spike when moving to a new disjoint chunk and the spikes are higher with higher severity of data distribution shift. As shown in Figure 14 in Appendix, the average value of the spike is about 7.1 for instance incremental sequence, 7.2 for random class incremental sequence, and 7.6 for distant class incremental sequence. See Appendix B.5 for more details on the pre-training loss.

---

> > > ### Author Response · Authors · 2021-11-23
> > > **Response to Reviewer 4UNG**
> > >
> > > > **Q5**: I am curious why there is a drop in SSL-ST wrt SSL-JT in figure 11 for instance incremental sequence setting. In Appendix B (Pretraining), it says that ‘we consider one random sequence as the data are randomly divided’. Does this mean that the way SSL-ST is implemented in an instance incremental setting, it is identical to SSL-JT? Please correct me if I am wrong. But if this is correct, shouldn’t the two curves have no performance difference?
> > >
> > > **A5**: Yes, SSL-ST shares the same data sequence as SSL-JT. Concretely, as for the instance incremental sequence, we randomly split ImageNet-1k, which is class-balanced, into 4 disjoint chunks. For all types of streaming data, once finishing the initial chunks split, we fix the data sequence for all corresponding sequential training or joint training practice. As for the question of why SSL-ST underperforms SSL-JT on the instance incremental sequence setting, this phenomenon is expected because SSL-JT uses all previous chunks while SSL-ST only uses the current chunk. Although the class distribution shift between chunks is negligible, more chunks mean more diverse samples/instances, which is beneficial for good representation learning, as proven by the ascending average accuracy with more chunks in Figure 9 in Appendix. In addition to that SSL-ST is worse than SSL-JT, we guess another observation would interest you, i.e. the performance gap between SL-ST and SL-JT is visibly larger than that of SSL, which has already interested Reviewer PFFi in Q6. This nontrivial observation further demonstrates that sequential SSL has less forgetting and better transfer ability than sequential SL, even on the nearly IID streaming data.
> > >
> > > &nbsp;
> > >
> > > [1] On Large-Batch Training for Deep Learning: Generalization Gap and Sharp Minima, ICLR 2016
> > >
> > > [2] Smoothout: Smoothing out sharp minima to improve generalization in deep learning, arxiv preprint, arxiv 2018
> > >
> > > [3] Understanding contrastive representation learning through alignment and uniformity on the hypersphere, ICML 2020

---

> > > > ### Comment · Reviewer_4UNG · 2021-11-30
> > > > **Response to authors**
> > > >
> > > > I acknowledge that I have read all the responses by the authors. I would like to thank the authors for putting in the effort to address the concerns and respond accordingly. After having gone through the responses, my comments are as follows,
> > > >
> > > > 1. The comparison w.r.t sharpness of loss minima between SSL and SL for different settings is insightful and helps to shed some light on the potential reason why SSL-ST and SSL-JT are closer in performance to each other compared to SL-ST and SL-JT. I do have some concerns in terms of how good the proxy is in assessing across different settings. From Table 4, we can see that the metric is better for SSL than for SL for all settings. But it seems that across different settings for SSL, instance incremental has the worst metric value compared to random class incremental and distant class incremental. This seems to contradict the inferences drawn from transfer learning experiments. Moreover, I am also not fully sure how good a metric sharpess of loss minima is for assessing forgetting. From the cited reference, it seems more to do with how well a model can generalize and not how well a model can learn incrementally over a stream of incoming data. I am not sure if the two concepts are interchangeable.
> > > >
> > > > 2. Looking at the training curves provided in Fig 14, I am doubtful if the small increase in the loss value can be considered a spike due to distribution drift when training starts for a new chunk. It would have been useful to also see the training curves for SSL-JT for a fair comparison. As also acknowledged by the authors, in the instance incremental setting, SSL-ST and SSL-JT share the same data sequence. There also we can observe that there is a rise in the loss with each new chunk. I believe this small rise in the loss is because of the queue being refreshed and requiring MoCo queue to refill and recalibrate the loss magnitude. Compared to instance incremental setting and the initial loss for first chunk, the initial loss for subsequent chunks for random class and distant class incremental seem comparable. Another peculiar observation is that in all settings across all chunks, the loss reaches the lowest value of ~6.6. This raises some concerns about the soundness of training performed.
> > > >
> > > > Given the above comments, I would like to raise my initial rating to 'marginally above acceptance threshold'. I feel that the new analysis does shed some light on why SSL-ST perform comparably to SSL-JT. The work also introduces some new perspective of comparing SSL and SL. However, I have some concerns on whether the analysis can be concretely used to justify the trends observed and whether the problem of continual learning and catastrophic forgetting makes sense for SSL because at large enough scale where SSL does have an impact, the practical approach to performing SSL is already using streaming data.

---

> > > > > ### Author Response · Authors · 2021-11-30
> > > > > **Response to Reviewer 4UNG**
> > > > >
> > > > > Thank you so much for raising the score. Thanks for reading our revision and responses thoroughly and providing precious feedback.  We carefully address your concerns as follows:
> > > > >
> > > > > > **Q6**: I do have some concerns in terms of how good the proxy is in assessing across different settings. From Table 4, we can see that the metric is better for SSL than for SL for all settings. But it seems that across different settings for SSL, instance incremental has the worst metric value compared to random class incremental and distant class incremental. This seems to contradict the inferences drawn from transfer learning experiments.
> > > > >
> > > > > **A6**: We want to provide more details about the experiments on the considered sharpness metric. We need to clarify that we cannot make comparisons of sharpness values across different settings. Because across different settings, the loss is calculated on different data samples and the classification tasks are different like the number of classes to be classified. Therefore, we can only compare the sharpness value between SL and SSL models trained and tested on the same streaming data.
> > > > >
> > > > > For all of the three ImageNet-based data sequences, we choose respective chunk A to pre-train a representation model. Then we evaluate the sharpness of the model following Eq. (1)-(2). Concretely, for the instance incremental sequence, chunk A contains IID samples from ImageNet-1K, classified into 1000 classes. For the random class incremental sequence, chunk A contains 250 random classes from ImageNet-1K, classified into 250 classes. For the distant class incremental sequence, chunk A contains 250 semantically similar classes from ImageNet-1K, classified into 250 classes.
> > > > > Although we cannot directly make across different settings, we can explain "why instance incremental has the worst metric value compared to random class incremental and distant class incremental." The classification task with 1,000 classes on instance incremental data is more challenging than both classification tasks with only 250 classes on the other two types of streaming data. Therefore, the sharpness value of instance incremental data is the highest. Different from the sharpness experiments, transfer learning experiments evaluate the performance of models on the same task. Therefore, there is no contradiction between the observations from both experiments. We will add more details about the sharpness experiments in our final version.
> > > > >
> > > > > &nbsp;
> > > > >
> > > > > ----------
> > > > >
> > > > > > **Q7**: Moreover, I am also not fully sure how good a metric sharpness of loss minima is for assessing forgetting. From the cited reference, it seems more to do with how well a model can generalize and not how well a model can learn incrementally over a stream of incoming data. I am not sure if the two concepts are interchangeable.
> > > > >
> > > > > **A7**: We are unaware of theoretical analysis on the relationship between the sharpness of minima in the loss landscape and forgetting in previous works.
> > > > > In this paper, we provide an intuitive explanation for this, similar to [4]. To be specific, when starting with flat minima in the loss landscape, we can expect that learning new data chunks will have a minor effect on the performance of previous chunks, as escaping the wide basin is difficult.
> > > > > We hope that our work inspires future work, particularly theoretical research, to shed more understanding of this critical problem.

---

> > > > > > ### Author Response · Authors · 2021-11-30
> > > > > > **Response to Reviewer 4UNG**
> > > > > >
> > > > > > > **Q8**: Looking at the training curves provided in Fig 14, I am doubtful if the small increase in the loss value can be considered a spike due to distribution drift when training starts for a new chunk. It would have been useful to also see the training curves for SSL-JT for a fair comparison. As also acknowledged by the authors, in the instance incremental setting, SSL-ST and SSL-JT share the same data sequence. There also we can observe that there is a rise in the loss with each new chunk. I believe this small rise in the loss is because of the queue being refreshed and requiring MoCo queue to refill and recalibrate the loss magnitude. Compared to instance incremental setting and the initial loss for first chunk, the initial loss for subsequent chunks for random class and distant class incremental seem comparable. Another peculiar observation is that in all settings across all chunks, the loss reaches the lowest value of ~6.6. This raises some concerns about the soundness of training performed.
> > > > > >
> > > > > > **A8**: The loss curves in Figure 14 illustrate the average MoCo-v2 training loss for each data chunk, i.e., the contrastive loss. The queue size is 65,536 for MoCo-v2, and the training batch size is 256. We record the training loss value averaged across one training batch and plot the loss data point every 250 training iterations, which means that after the first point in the loss curve, we have a filled queue for contrastive learning. Therefore, the loss spike is not because of the training mechanism, including the queue's refreshing in early training iterations. Instead, it is reasonable to ascribe the loss spike to the distribution shift change of training data, i.e., new chunk.
> > > > > > As for the first chunk, the initial average contrastive loss is significant (around 10) because we randomly initialize the representation model. As for subsequent chunks, we inherit the representation model from the previous step. Therefore the initial average contrastive loss is not significant, i.e., around 7.2.
> > > > > > On each chunk, we train the model for 200 epochs and finally observe the convergence of contrastive loss. We find that the average contrastive loss would generally converge to the value of around 6.6 for all chunks. We adopt the same training mechanism for each chunk, including the difficulty of the instance discrimination task and hyperparameter settings. Additionally, we perform contrastive learning until convergence on each chunk. As a result, the final converged loss value is similar for different chunks across different settings.
> > > > > >
> > > > > > We are open to any further discussion. Also, we will add all the relevant details discussed above into our final version.
> > > > > >
> > > > > > [4] Linear mode connectivity in multitask and continual learning. ICLR’20.

---

### Author Response · Authors · 2021-11-23
**General Response to all reviewers**

We thank all reviewers for their time and valuable feedback. In this work,  we appreciate reviewers acknowledge that:
* Our work “firstly investigates the sequential training situation for self-supervised learning”(Reviewer QZfY), and “the task studied in the manuscript is highly practical”(Reviewer RJbi) and “important”(Reviewer PFFi).
* We provide “an extensive framework” which “unifies the different settings considered by previous works in this domain and provides a comprehensive benchmark”(Reviewer 4UNG)
* Our empirical study is “thorough” and “gives interesting observations”(Reviewer PFFi). We conduct experiments on “four categories of incremental learning setup”(Reviewer RJbi), which are “interesting and to the point”(Reviewer RJbi).
* Our analysis is “particularly insightful”. Especially, the analysis on “space-time efficiency and backward/forward transfer” is “quite extensive and useful to understand the benefits”(Reviewer 4UNG).

Then in our reply to each reviewer, we address each reviewer’s detailed concerns point by point. **Note that, in revision, we introduce an explanation for why SSL forgets less than SL, which provides more insights for self-supervised learning and continual learning and further improves the novelty of our work. Specifically, we find that SSL models have wider minima than SL models in the loss landscape, which we hypothesize is the reason for less forgetting of SSL models.** We have revised the manuscript according to reviewers’ comments. The main changes(highlighted in blue) we made include:

1. In Sec. 1 Introduction, we add more introduction about our proposed streaming pre-training data. Specifically, we improve the clarification in Fig.1 and add more introduction for the four types of streaming data. We further clarify the four main contributions or takeaways of this paper.
2. In Sec. 2 Related works, we add more discussion about downstream-agnostic and downstream-aware pretraining.
3. In Sec 4.2 and Appendix B.3, we provide a more detailed explanation for possible negative transfer on the downstream tasks in learning with streaming data.
4. In Sec 5.1, we provide a more detailed introduction for backward and forward transfer. Moreover, in Appendix C.2, we present the calculation details for BWT/FWT.
5. In Sec 5.3, we add an in-depth explanation for why self-supervised continual learning suffers less forgetting,  from the perspective of the flatness of minima in the loss landscape. In Appendix C.4, we add more details and visualization for the flatness analysis.
6. In Appendix A.2, we add more details of MoCo-v2 pre-training.
7. In Appendix A.4, we add more details of few-shot evaluation.
8. In Appendix B.2, we add the results on the longer sequence (8) with a larger-scale dataset, ImageNet-21K.
9. In Appendix B.5, we provide the self-supervised training loss curve at each step for each type of streaming data.
10. In Appendix C.1, we add uniformity analysis for the representation.
11. We move the CKA analysis between ST and JT models to Appendix C.3.

We sincerely hope that this work would benefit both self-supervised learning and continual learning communities. We would really appreciate that the reviewers can check whether your concerns have been addressed. We are also open to answering any further questions about the revision and would take it into consideration in the final version.

---

### Author Response · Authors · 2021-11-29
**Follow-up discussion**

We again thank all reviewers for the detailed and valuable comments, which helped us complete the significant revisions to our submission. Since the discussion period will end in 2 days, we would like to know if there is any remaining question that we can resolve? We look forward to your response.

---

### Decision · Program_Chairs · 2022-01-20

**Decision:**

Accept (Poster)

**Comment:**

This paper performs a comprehensive investigation on self-supervised pre-training with streaming data. Reviewers agreed that the task studied in this paper is highly practical and important, and the analysis is insightful. Meanwhile, reviewers raised some concerns such as empirical setups and insights. In the revised paper, the authors provided more justifications and added more analysis such as few-shot evaluation and uniformity analysis. After the discussion period, most reviewers are positive about this paper.

Overall, I recommend to accept this paper. I encourage the authors to take the review feedback into account in the final version.